# Towards Prospective Medical Image Reconstruction via Knowledge-Informed Dynamic Optimal Transport

**Taoran Zheng[1],[*], Yan Yang[1],[*],[†], Xing Li[1], Xiang Gu[1],[†], Jian Sun[1],[2], Zongben Xu[1]**

[1]School of Mathematics and Statistics, Xi'an Jiaotong University, Xi'an, China
[2]State Industry-Education Integration Center for Medical Innovations at Xi'an Jiaotong University
`taoranzheng@stu.xjtu.edu.cn`; `{listar0810,yangyan,xianggu,jiansun,zbxu}@xjtu.edu.cn`

## Abstract

Medical image reconstruction from measurement data is a vital but challenging inverse problem. Deep learning approaches have achieved promising results, but often requires paired measurement and high-quality images, which is typically simulated through a forward model, i.e., retrospective reconstruction. However, training on simulated pairs commonly leads to performance degradation on real prospective data due to the retrospective-to-prospective gap caused by incomplete imaging knowledge in simulation. To address this challenge, this paper introduces imaging Knowledge-Informed Dynamic Optimal Transport (KIDOT), a novel dynamic optimal transport framework with optimality in the sense of preserving consistency with imaging physics in transport, that conceptualizes reconstruction as finding a dynamic transport path. KIDOT learns from unpaired data by modeling reconstruction as a continuous evolution path from measurements to images, guided by an imaging knowledge-informed cost function and transport equation. This dynamic and knowledge-aware approach enhances robustness and better leverages unpaired data while respecting acquisition physics. Theoretically, we demonstrate that KIDOT naturally generalizes dynamic optimal transport, ensuring its mathematical rationale and solution existence. Extensive experiments on MRI and CT reconstruction demonstrate KIDOT's superior performance. Code is available at https://github.com/TaoranZheng717/KIDOT.

## 1 Introduction

Medical imaging techniques like Magnetic Resonance Imaging (MRI) and Computed Tomography (CT) are vital tools for visualizing internal anatomy, yet reconstructing high-fidelity images from the acquired measurements remains a significant challenge [1, 2]. This process is fundamentally an ill-posed inverse problem inferring a complete, clean image from often incomplete and noisy measurement data. Classical reconstruction algorithms, such as filtered back-projection or iterative methods [3–7], typically rely on precise mathematical modeling of the image acquisition physics, incorporating essential imaging knowledge. However, accurately capturing complex factors present in real-world scenarios (e.g., noise, non-ideal sampling) within these models remains challenging. Consequently, the performance of traditional methods can degrade significantly when this knowledge is incomplete or imperfect, yielding images with noise or residual artifacts.

Deep learning (DL) has emerged as a powerful alternative [8–10], achieving state-of-the-art results by learning complex mappings directly from data. However, the predominant supervised DL paradigm[11–13] faces a critical bottleneck: the requirement for large datasets of perfectly registered pairs of low-quality input measurements and their corresponding high-quality ground truth images.

---

[*]These authors contributed equally. [†]Corresponding authors.

39th Conference on Neural Information Processing Systems (NeurIPS 2025).

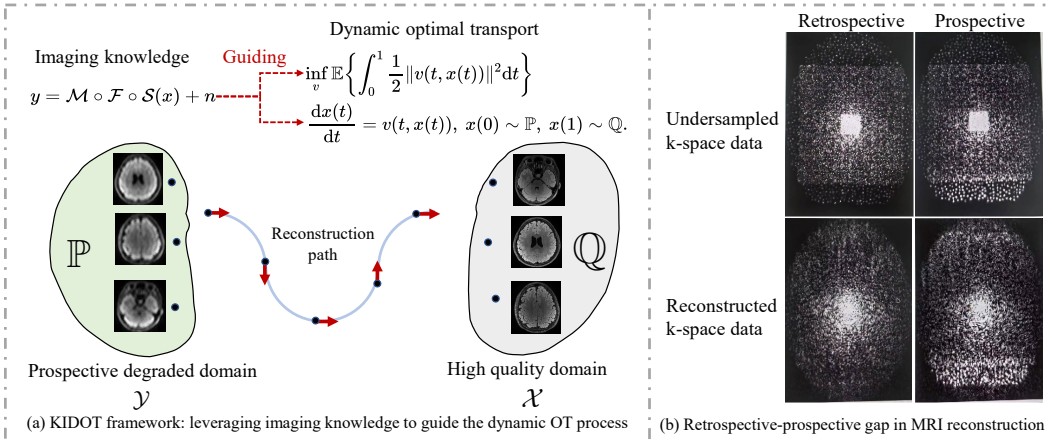

Figure 1: (a) Core concept of KIDOT: modeling image reconstruction as a continuous evolution from a prospective degraded distribution $\mathbb{P}$ to a high quality image distribution $\mathbb{Q}$, guided by dynamic OT. (b) The retrospective-prospective gap in MRI: a visual comparison of k-space data from retrospective simulations versus real prospective undersampling (top), and their corresponding k-space reconstructions via supervised learning (bottom).

Obtaining such perfectly aligned data in clinical practice is often difficult, costly, or ethically constrained. Furthermore, many real-world datasets inherently lack perfect pairing due to patient motion, or differences in acquisition protocols between low-quality and high-quality scans [14, 15].

To circumvent the lack of real paired data, a common practice [16, 17] is to generate simulated low-quality data using simplified imaging knowledge (e.g., adding modeled noise, applying idealized undersampling masks) from available high-quality images, creating artificial pairs for supervised training. However, a significant gap often exists between these simulations and real-world prospective measurements. Figure 1(b) shows examples of the retrospective-prospective gap in MRI, where k-space data from retrospective simulations (left) differ from that of prospective acquisitions (right) in sampling patterns and reconstructed high-frequency regions. Retrospective simulations often idealize the true data acquisition process, struggling to fully capture the complex noise and genuine artifact characteristics inherent in real-world prospective measurements. Models trained solely on such simulated pairs, therefore, often exhibit a performance drop when deployed in prospective clinical settings, highlighting their sensitivity to distribution shifts arising from the disparity between simulated and real-world imaging knowledge. Ideally, a reconstruction method should effectively leverage all available data resources, including potentially large amounts of unpaired real measurements and real high-quality images, while robustly incorporating reliable imaging knowledge. This motivates exploring frameworks capable of learning from unpaired distributions in a physics-aware manner. Optimal Transport (OT) [18, 19] offers a principled mathematical approach for transporting probability distributions or finding mappings between them, with the potential to learn from unpaired data. However, conventional applications of OT often adopt a static perspective, learning a direct map and potentially overlooking the dynamic physical processes inherent in image reconstruction.

To tackle the above challenges, this paper introduces imaging Knowledge-Informed Dynamic Optimal Transport (KIDOT), a novel framework for unpaired medical image reconstruction. KIDOT integrates imaging knowledge, primarily through the forward physical model, directly into a dynamic OT formulation. As conceptually depicted in Fig. 1(a), instead of learning a static transformation, KIDOT models reconstruction as a continuous evolution path from the measurement distribution ($\mathbb{P}$) to the target image distribution ($\mathbb{Q}$). This evolution is guided by an instantaneous cost function and governed by a transport equation, both incorporating imaging knowledge, ensuring the transport path remains consistent with the physics throughout the transformation. This dynamic, knowledge-informed approach enables more realistic modeling, suited for leveraging unpaired data while remaining grounded in the physics of image acquisition. To translate KIDOT into practice, we introduce a neural network implementation strategy that learns from a combination of unpaired real and paired simulated medical images. Furthermore, the theoretical underpinnings of KIDOT, including the rationale and the existence of its learned transport solution, are established, validating its mathematical rigor.

The main contributions of this work are threefold: 1) We propose the KIDOT framework, which uniquely integrates imaging physics into both the cost function and transport equation of dynamic OT for prospective medical image reconstruction. To the best of our knowledge, this is the first dynamic OT framework that integrates medical imaging physics, with optimality in the sense of preserving consistency with imaging knowledge throughout its transport path. 2) We develop a practical implementation algorithm of KIDOT based on neural networks, enabling it to learn to reconstruct from unpaired medical image data. Theoretical analysis for KIDOT is established, showing its rationale and the existence of the learned transport solution. 3) We demonstrate the practical effectiveness and superior performance of KIDOT through extensive experiments on challenging MRI and CT reconstruction tasks, particularly showcasing its advantages in handling prospectively acquired and unpaired clinical data.

## 2    Background

**Prospective medical image reconstruction.**    Medical imaging seeks to reconstruct internal structures from indirect, often noisy measurements. This task is fundamentally an ill-posed inverse problem: determining an underlying image $x$ given observed data $y$. The relationship is typically modeled as $y = \mathcal{A}(x) + n$, where $\mathcal{A}$ represents the forward model dictated by the imaging physics (e.g., the Radon transform in CT [20] or the partial Fourier transform in MRI [21]), and $n$ accounts for measurement noise and errors. Current approaches to solving such inverse problems can be broadly categorized into three types. Classical iterative methods [3, 4] minimize the difference between predicted and measured data, guided by the physical model $\mathcal{A}$ and incorporating prior assumptions about the image $x$ (e.g., sparsity [3] or low-rank structure [4]) via regularization:

$$\hat{x} = \arg \min_x \|\mathcal{A}(x) - y\|^2 + \lambda \mathcal{R}(x), \tag{1}$$

where $\mathcal{R}(x)$ is the regularization term, and $\lambda$ is the trade-off parameter. While grounded in physics, these methods may struggle to recover fine details from limited or noisy data. More recently, deep learning techniques [22–26] have shown promise, learning direct mappings from measurements $y$ to images $x$. These data-driven models can achieve high accuracy and speed but typically require large datasets of paired examples (i.e., measurement $y$ and corresponding ground truth image $x$). Hybrid approaches [11, 27] attempt to combine the strengths of both data- and model-driven methods, integrating physical models within neural network architectures. However, like purely data-driven methods, they usually depend on the availability of paired training data. A significant practical challenge arises because acquiring such perfectly aligned, high-quality reference images alongside clinical measurements is often difficult or impossible (e.g., obtaining perfectly matched low-dose and standard-dose CT scans). This common scenario, where only measurements $y$ are readily available without corresponding high-quality $x$, defines the challenging prospective inverse problem setting that remains underexplored [16, 17].

**Optimal transport and its applications in medical image reconstruction.**    OT offers a robust mathematical framework for comparing probability distributions via the minimal cost required to transform one into another [18, 28, 29]. Static OT finds an optimal coupling $\pi$ between source $\mathbb{P}$ and target $\mathbb{Q}$ distributions that minimizes the expected transport cost [18]:

$$\inf_{\pi \in \Pi(\mathbb{P}, \mathbb{Q})} \mathbb{E}_{(x,y) \sim \pi}[c(x, y)], \tag{2}$$

where $c$ is the cost function, and $\Pi$ is the set of joint distributions with marginals $\mathbb{P}, \mathbb{Q}$. Extending this, dynamic OT models the continuous evolution from $\mathbb{P}$ to $\mathbb{Q}$ over time [18, 19]. Dynamic OT seeks a velocity field $v(t, x(t))$ that optimally transports particles from an initial distribution $x(0) \sim \mathbb{P}$ to a final distribution $x(1) \sim \mathbb{Q}$. This is achieved by finding $v$ that minimizes the action functional:

$$\inf_v \mathbb{E} \left\{ \int_0^1 \frac{1}{2} \|v(t, x(t))\|^2 \mathrm{d}t \right\} \quad \text{s.t.} \ \frac{\mathrm{d}x(t)}{\mathrm{d}t} = v(t, x(t)), \ x(0) \sim \mathbb{P}, \ x(1) \sim \mathbb{Q}. \tag{3}$$

The OT principles have proven valuable for inverse problems lacking paired data, e.g., natural image processing [30–34]. In medical imaging, static OT has been explored to address the challenges of unpaired or misaligned clinical data. For example, [35, 36] use static OT to derive a CycleGAN framework for learning from unpaired data. [37] further integrates variational methods with reconstruction

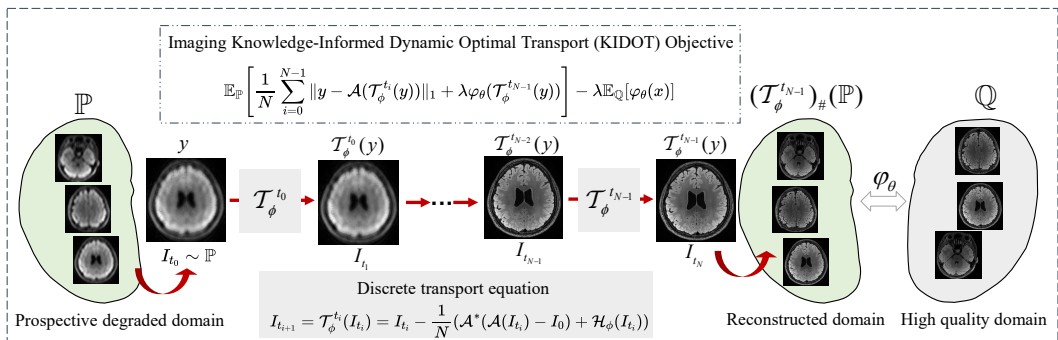

Figure 2: Illustration of the KIDOT framework. KIDOT employs a sequence of transformations $T_\phi^{t_i}$, governed by a learned transport equation driven by the medical image reconstruction model, to map measurement $y$ from the prospective degraded domain to the reconstructed domain. The transport process is guided by the KIDOT objective function, which critically incorporates imaging knowledge by enforcing data consistency along the transport path. Simultaneously, it promotes distribution matching between the reconstructed domain and the target high-quality domain using potentials $\varphi_\theta$.

and Wasserstein-1 distance for unpaired learning in CT. [38, 39] preserve structural consistency [38] or incorporate contextual features [39] in OT-based generative models. Static OT has also been utilized for aligning and synthesizing different MRI sequences to enhance reconstruction quality [40]. More recently, dynamic OT has been explored for semi-supervised problems, FSBM [41] utilizes a distance-preserving objective to build a dynamic transport process in Schrödinger bridge framework.

Despite these advances, prior OT-based methods in medical imaging often utilize generic costs (e.g., Euclidean distance) disconnected from the underlying image formation physics, or employ static OT which considers only the source and target distributions, without modeling the dynamics of the transformation between them. We introduce a novel constraint and objective by requiring the dynamic OT path to be consistent with the physical forward model of the imaging system. This embeds crucial domain knowledge directly into the transport dynamics, offering an imaging-informed alternative to standard dynamic OT formulations for inverse problems.

## 3 Imaging Knowledge-Informed Dynamic Optimal Transport

Given measurement distribution $\mathbb{P}$ and high quality image distribution $\mathbb{Q}$ in prospective medical image reconstruction, this paper aims to model a principled evolution from an initial state $y \sim \mathbb{P}$ to corresponding target state $x \sim \mathbb{Q}$. Dynamic OT provides a framework for modeling the evolution between probability distributions, characterized by a transport cost and governing transport dynamics. Conventional dynamic OT often assumes the transport cost is based on a standard metric distance (e.g., Euclidean distance), optimizing for the shortest path length. However, in inverse problems, e.g., medical image reconstruction, a more physically meaningful cost might relate the evolving estimate to the observed measurements via the forward model, and this cost may not be a standard metric. Meanwhile, the physical knowledge should also be incorporated into the transport dynamics. This necessitates adapting the dynamic OT framework. Towards this goal, we propose an imaging Knowledge-Informed Dynamic Optimal Transport (KIDOT) approach for prospective medical image reconstruction. As illustrated in Fig. 2, in KIDOT, we design a transport cost reflecting local consistency with the imaging physics at each point along the evolution. Meanwhile, the dynamics of KIDOT, represented by the transport equation, is driven by the gradient flow of inverse problems that incorporate imaging knowledge. We also develop a practical implementation based on neural networks for KIDOT, enabling it to learn from unpaired real measurements for prospective medical image reconstruction. Next, we elaborate on these components in detail.

### 3.1 Dynamic Transport Cost Informed by Imaging Knowledge

In static OT, the objective is to find a coupling $\pi$ between source $\mathbb{P}$ and target $\mathbb{Q}$ distributions that minimizes an average transport cost (Eq. (2)). To effectively apply OT to imaging inverse problems, where measurements $y$ are related to an underlying image $x$ by a forward model $y \approx \mathcal{A}(x)$, the

cost function $c(x, y)$ should encode this physical relationship. Therefore, instead of using generic distances, we propose an imaging-informed cost based on data fidelity: $c(x, y) = \|y - \mathcal{A}(x)\|_1$. This cost directly quantifies the discrepancy between an observed measurement $y$ and the measurement predicted by the forward model $\mathcal{A}$ on a candidate image $x$.

Yet, how to define the transport cost in the dynamic setting is difficult. A key challenge arises because our imaging-informed cost $c(x, y) = \|y - \mathcal{A}(x)\|_1$ is generally not a metric. Consequently, the standard dynamic OT formulation based on minimizing path length defined based on a metric is unsuitable. As an alternative, we propose to minimize the expected value of an integrated instantaneous cost over potential paths:

$$\inf_v \mathbb{E}_\mathbb{P} \left[ \int_0^1 c(I_t, I_0) \, \mathrm{d}t \right], \quad \text{where } c(x, y) = \|y - \mathcal{A}(x)\|_1. \tag{4}$$

Here, the dynamics $\frac{\mathrm{d}I_t}{\mathrm{d}t} = v(t, I_t)$ are governed by the velocity field $v$, and $I_t$ represents the path evolving from an initial measurement $I_0$. This objective seeks dynamics $v$ such that, on average, the evolving path $I_t$ maintains consistency with its specific originating measurement $I_0$ (via the forward model $\mathcal{A}$) throughout the entire time interval $[0, 1]$. We next analyze the rationale of this cost.

**Rationale analysis.** To evaluate the suitability of minimizing such an integrated cost, we analyze its behavior in the Euclidean space. Considering paths connecting a starting point $y$ to an endpoint $x$, we examine which path minimizes the integral of the instantaneous Euclidean distance between the point on the path $s(t)$ and the fixed endpoint $x$. We have the following theorem.

**Theorem 3.1** (Existence and Geometry of Minimizer for $L^2$ Distance Integral). *Let $x, y \in \mathbb{R}^n$ be distinct points and let $M \geq \|x - y\|_2$ be a constant. Define the feasible set of paths $\mathcal{X}_M$ as $\mathcal{X}_M = \{s \in AC([0, 1], \mathbb{R}^n) \mid s(0) = y, s(1) = x, \|s'\|_\infty \leq M\}$, where $AC([0, 1], \mathbb{R}^n)$ is the space of absolutely continuous paths. Then, an optimal solution $s^* \in \mathcal{X}_M$ to problem*

$$\inf_{s \in \mathcal{X}_M} \int_0^1 \|s(t) - y\|_2^2 dt \tag{5}$$

*exists and its geometric trajectory $\{s^*(t) \mid t \in [0, 1]\}$ is the straight line segment connecting $y$ and $x$.*

The proof is provided in Appendix A.2. This theorem indicates that when the static cost is taken as the squared Euclidean distance, minimizing the proposed dynamic transport cost recovers the standard straight-line path, consistent with the conventional dynamic OT. This suggests that minimizing this integral formulation could be a reasonable way to define transport dynamics based on general cost, including our non-metric, imaging knowledge-informed data-fidelity cost.

### 3.2 Transport Equation Guided by Imaging Knowledge

We now specify the dynamics that governs the transformation path $I_t$ from measurements to reconstructed images, guided by imaging knowledge. Our approach draws inspiration from inverse problems, which commonly minimize an objective that combines data consistency with a regularization term $\mathcal{R}(I)$ encoding prior knowledge about the desired image. The continuous-time gradient flows of the inverse problems naturally provide a principled evolution dynamic. Specifically, minimizing Eq. (1) via gradient flow leads to the following ordinary differential equation (ODE):

$$\frac{\mathrm{d}I_t}{\mathrm{d}t} = -(\mathcal{A}^*(\mathcal{A}(I_t) - I_0) + \nabla\mathcal{R}(I_t)), \tag{6}$$

where $I_0$ represents the initial measurement $y$, $\mathcal{A}^*$ is the adjoint of the forward operator $\mathcal{A}$, and $\nabla\mathcal{R}(I_t)$ is the gradient of the regularization term evaluated at the current state $I_t$. This ODE describes a path moving away from simple measurements towards reconstructions that better satisfy both data fidelity and prior constraints. We adopt this gradient flow to define the velocity field $v(t, I_t)$ that drives the transport in our KIDOT framework. Note that the regularization $\mathcal{R}$ is difficult to define and is often learned based on paired data in a supervised manner in previous methods. By taking Eq. (6) as the transport equation, our KIDOT will provide an approach to learn the regularization/prior from prospective unpaired data.

Combining the proposed integrated cost (Eq. (4)) with the imaging-knowledge-guided dynamics (Eq. (6)), the KIDOT formulation seeks the optimal regularization $\mathcal{R}$ by solving:

$$\inf_{\mathcal{R}} \mathbb{E}_{I_0 \sim \mathbb{P}} \left[ \int_0^1 \| I_0 - \mathcal{A}(I_t) \|_1 \, \mathrm{d}t \right]$$

$$\text{s.t. } \frac{\mathrm{d}I_t}{\mathrm{d}t} = -(\mathcal{A}^*(\mathcal{A}(I_t) - I_0) + \nabla \mathcal{R}(I_t)), \ I_0 \sim \mathbb{P}, I_1 \sim \mathbb{Q}. \tag{7}$$

In problem (7), the objective function minimizes the expected cost along the trajectory, while the constraints enforce that the evolution follows the defined dynamics that starts from the distribution of input measurements $\mathbb{P}$, and ultimately transforms it into the target distribution of high-quality images $\mathbb{Q}$ at time $t = 1$. This formulation integrates imaging knowledge into both the transport cost and equation, forming an imaging knowledge-informed dynamic OT model. We note that while the gradient flow in Eq. (6) serves as our main example, the KIDOT framework is inherently flexible and can readily accommodate more advanced optimization flows, e.g., proximal gradient flows, to incorporate the imaging knowledge into the transport equation in more different ways.

### 3.3 Practical Implementation Based on Neural Networks

This section focuses on the practical solution to KIDOT problem (7). Since the optimization variable is a function in the KIDOT problem, we parameterize it using neural networks to ease implementation.

**Parametrization and relaxation.** For convenience, we directly parameterize $\nabla \mathcal{R}$ using a neural network $\mathcal{H}_\phi$ with parameters $\phi$. To solve problem (7), another challenge is the ODE-based constraint with initial and ending distribution constraints, making optimization difficult. To tackle this challenge, we apply a relaxation to the ending distribution to seek an approximation of the optimal solution. Specifically, given an initial state $I_0 \sim \mathbb{P}$, we produce $I_t$ using ODE by $I_t \triangleq \mathcal{T}_\phi^t(I_0) = I_0 - \int_0^t [\mathcal{A}^*(\mathcal{A}(I_\tau) - I_0) + \mathcal{H}_\phi(I_\tau)] \mathrm{d}\tau$.

By denoting the distribution of $\mathcal{T}_\phi^1(I_0)$ as $\mathcal{T}_{\phi\#}^1(\mathbb{P})$, the constraint $I_1 \sim \mathbb{Q}$ becomes $\mathcal{T}_{\phi\#}^1(\mathbb{P}) = \mathbb{Q}$, equivalent to $W_1(\mathcal{T}_{\phi\#}^1(\mathbb{P}), \mathbb{Q}) = 0$ where $W_1$ is the 1-Wasserstein distance. By introducing the Lagrange relaxation, problem (7) becomes

$$\inf_\phi J(\phi | \lambda, \mathbb{P}, \mathbb{Q}) \triangleq \mathcal{C}(\phi) + \lambda W_1(\mathcal{T}_{\phi\#}^1(\mathbb{P}), \mathbb{Q}), \tag{8}$$

where $\mathcal{C}(\phi) = \mathbb{E}_{I_0 \sim \mathbb{P}} \left[ \int_0^1 c(\mathcal{T}_\phi^t(I_0), I_0) \, \mathrm{d}t \right]$. Leveraging the Kantorovich-Rubinstein duality [18] for the $W_1$ distance, we further transform the optimization problem to

$$\inf_\phi \left\{ \mathcal{C}(\phi) + \lambda \sup_{\|\varphi\|_{\mathrm{Lip}} \leq 1} \left( \mathbb{E}_{x \sim \mathbb{Q}}[\varphi(x)] - \mathbb{E}_{I_0 \sim \mathbb{P}}[\varphi(\mathcal{T}_\phi^1(I_0))] \right) \right\}, \tag{9}$$

where $\| \cdot \|_{\mathrm{Lip}}$ is the Lipschitz norm.

**Discretization and training.** To solve the above problem, we also parameterize $\varphi$ using a neural network $\varphi_\theta$ with parameters $\theta$. The integral to produce $\mathcal{T}_\phi^t(I_0)$ and the cost are approximated numerically via a discrete sum over $N$ time steps $0 = t_0 < t_1 < \cdots < t_N = 1$, with a step size $t_{i+1} - t_i = 1/N$. Using a forward Euler discretization for the ODE in Eq. (7), the evolution at each discrete step becomes $I_{t_{i+1}} = I_{t_i} - \frac{1}{N} \left( \mathcal{A}^*(\mathcal{A}(I_{t_i}) - I_0) + \mathcal{H}_\phi(I_{t_i}) \right)$, where $I_{t_0} = I_0$. Let $\mathcal{T}_\phi^{t_i}(I_0)$ denote the state $I_{t_{i+1}}$ obtained after $i + 1$ such steps. Substituting these into the dual objective Eq. (9) and discretizing the time integral, we arrive at the following loss function for training:

$$\mathcal{L}_{\mathrm{KIDOT}}(\phi, \theta) = \mathbb{E}_\mathbb{P} \left[ \frac{1}{N} \sum_{i=0}^{N-1} \| y - \mathcal{A}(\mathcal{T}_\phi^{t_i}(y)) \|_1 + \lambda \varphi_\theta(\mathcal{T}_\phi^{t_{N-1}}(y)) \right] - \lambda \mathbb{E}_\mathbb{Q}[\varphi_\theta(x)]. \tag{10}$$

$\mathcal{L}_{\mathrm{KIDOT}}$ can be implemented on unpaired mini-batch samples of the unpaired prospective medical images. In medical image reconstruction, the simulation measurements produced by the forward model $\mathcal{A}$ on high-quality medical images are also available for training, serving as partially paired data $P_{pair}$. For these data, we apply a supervised training loss as

$$\mathcal{L}_{\mathrm{SUP}}(\phi) = \mathbb{E}_{(y_p, x_p) \sim P_{pair}}[\mathcal{F}(\mathcal{T}_\phi^{t_{N-1}}(y_p), x_p)], \tag{11}$$

where $\mathcal{F}(\cdot, \cdot)$ is a standard supervised loss function (e.g., $L_1$ or $L_2$ distance), encouraging the predicted endpoint $\mathcal{T}_\phi^{t_{N-1}}(y_p)$ to match the known ground truth $x_p$. The final training objective is

$$\inf_\phi \sup_\theta \mathcal{L}_{\text{KIDOT}}(\phi, \theta) + \gamma \mathcal{L}_{\text{SUP}}(\phi), \tag{12}$$

where $\gamma$ is a hyperparameter. Eq. (12) is optimized by alternately updating $\phi$ and $\theta$ through gradient descent and ascent, respectively. We provide the detailed training algorithm in Appendix A.1.

**Theoretical analysis.** We now study the existence of an optimal solution to the formulation presented in Eq. (8). We make the assumptions: A1) Measures $\mathbb{P}$ and $\mathbb{Q}$ have compact support, and the parameter set $K \subset \mathbb{R}^d$ is non-empty and compact. A2) $\mathcal{T}_{\phi_n}^t \to \mathcal{T}_\phi^t$ pointwise whenever $\phi_n \to \phi$ and there exists $h \in L^1([0, 1])$ such that $|c(\mathcal{T}_{\phi_n}^t(y), y)| \leq h(t)$ for all $n$ and $t \in [0, 1]$. A3) $\sup_{\phi \in K} \|\mathcal{T}_\phi^t\|_\infty < \infty$. A4) For fixed $y$, the cost function $c(y, \cdot)$ is continuous in its second argument.

**Theorem 3.2** (Existence of Minimizer for KIDOT Objective). *Suppose the assumptions A1-A4 hold, then there exists a minimizer for the optimization problem* $\inf_{\phi \in K} J(\phi | \lambda, \mathbb{P}, \mathbb{Q})$.

The proof is provided in Appendix A.3. This theorem confirms that the KIDOT objective is well-defined and that a set of optimal parameters $\phi^*$ guiding the transport path exists.

## 4 Experiments

We conduct experiments encompassed simulated MRI data, prospectively acquired real-world MRI data, and prospectively acquired clinical CT data. Full experimental details and supplementary results are available in Appendix B due to space limit.

**Simulated MRI data.** Our experiments on simulated MRI data utilized the publicly accessible fastMRI multi-coil knee dataset [42]. We selected 2500 fully sampled MRI slices, partitioning them into 1000 for training, 500 for validation, and 1000 for testing. Undersampled measurements were simulated using k-space masks corresponding to a 4x acceleration factor. A key aspect of this evaluation was to assess KIDOT's robustness to scenarios where test data characteristics differ from training data, a common challenge in prospective clinical deployment. To achieve this, we generated two distinct sets of undersampled inputs: one set, with specific sampling patterns, was used for training supervised baselines, while a different set of sampling patterns was employed for testing to mimic prospective acquisition conditions. For this dataset, the core transport network $\mathcal{T}_\phi$ within KIDOT was implemented using an unfolding architecture based on E2E-VarNet [12].

Quantitative results on the simulated MRI dataset are presented in Table 1. KIDOT demonstrates superior performance across all evaluated metrics, encompassing both image fidelity measures (PSNR and SSIM) and perceptual quality indicators (FID [43] and KID [44]). These scores outperform conventional methods like CS-wavelet [45, 46], supervised learning models like E2E-VarNet [12], contemporary unpaired learning techniques such as OT-CycleGAN [35], UAR [37] and FSBM [41], as well as strong diffusion-based models like DDS [47]. This highlights KIDOT's effectiveness in producing reconstructions that achieve both high fidelity to the ground truth and strong perceptual realism, even under these challenging simulated prospective conditions. Visual comparisons, further illustrating these improvements, are provided in Appendix B.8.

Table 1: Quantitative comparison on the **simulated MRI dataset** (4x acceleration). Metrics include PSNR, SSIM (higher is better $\uparrow$), FID, and KID$\times$100 (lower is better $\downarrow$). Best results are **bolded**.

| Method | PSNR $\uparrow$ | SSIM $\uparrow$ | FID $\downarrow$ | KID$\times$100 $\downarrow$ |
|---|---|---|---|---|
| CS-wavelet [45] | 29.17 | 0.7981 | 95.76 | 5.77 |
| E2E-VarNet [12] | 32.56 | 0.8372 | 24.35 | 1.29 |
| OT-CycleGAN [35] | 26.45 | 0.7751 | 156.43 | 7.25 |
| UAR [37] | 33.01 | 0.8425 | 19.56 | 1.05 |
| FSBM [41] | 26.52 | 0.7634 | 113.62 | 6.03 |
| DDS [47] | 31.27 | 0.8136 | 29.46 | 1.65 |
| **KIDOT (ours)** | **33.31** | **0.8518** | **10.68** | **0.83** |

**Real prospective MRI data.** To further assess KIDOT's performance in realistic clinical scenarios, we utilized a multi-contrast brain MRI dataset acquired prospectively using a 3.0T United Imaging scanner equipped with a 32-channel head coil. Our experiments centered on the T2-FLAIR weighted sequence, which was undersampled with a 10x acceleration factor. The dataset, comprising images of size $256 \times 256 \times 176$, was divided into 3077 training, 1860 validation, and 3077 test slices. A key challenge with this dataset is the separate acquisition of undersampled data and their corresponding fully sampled references, a common practice that often introduces inherent spatial misalignments between them. Consequently, standard pixel-wise fidelity metrics such as PSNR or SSIM are less reliable for evaluating performance on the test set.

For this dataset, the transport network $\mathcal{T}_\phi$ in KIDOT was parameterized using an unfolding architecture based on PromptMR [13]. Given the inherent spatial misalignments, which render pixel-wise metrics unreliable, our evaluation on this dataset focused on distribution-based metrics: FID and KID. Furthermore, we conducted a rigorous benchmark against a state-of-the-art DDS model under three practical strategies: using a pre-trained model, fine-tuning it on our data, and training it from scratch. As detailed in Table 2, KIDOT achieved the leading FID and KID scores, outperforming both supervised and unsupervised baselines. KIDOT surpasses all DDS variants, which highlights a key advantage: its physics-informed dynamics provide a powerful inductive bias, making it more data-efficient and robust, especially when real training data is limited. Visual results presented in Figure 3 corroborates these quantitative findings. On representative test slices, KIDOT reconstructions exhibit clear anatomical detail, effective noise suppression, and reduced Gibbs artifacts. Particularly in challenging areas (e.g., highlighted region in red), KIDOT demonstrates superior preservation of fine structures compared to other approaches. Notably, as FSBM is not tailored for medical imaging and its visual results were poor, we omitted its visualizations for brevity.

Table 2: Quantitative comparison on the **Real Prospective MRI Data** ($10\times$ acceleration). Metrics include FID and KID$\times100$ (lower is better $\downarrow$). Best results are **bolded**.

| Method | FID $\downarrow$ | KID$\times100 \downarrow$ |
|---|---|---|
| CS-wavelet [45] | 67.75 | 3.34 |
| E2E-VarNet [12] | 65.27 | 2.83 |
| PromptMR [13] | 29.52 | 1.46 |
| OT-CycleGAN [35] | 221.83 | 10.94 |
| UAR [37] | 29.09 | 1.23 |
| FSBM [41] | 147.62 | 7.13 |
| DDS (Pre-trained) [47] | 62.94 | 2.71 |
| DDS (From scratch) [47] | 58.32 | 2.53 |
| DDS (Fine-tuned) [47] | 39.78 | 1.84 |
| **KIDOT (ours)** | **28.26** | **1.12** |

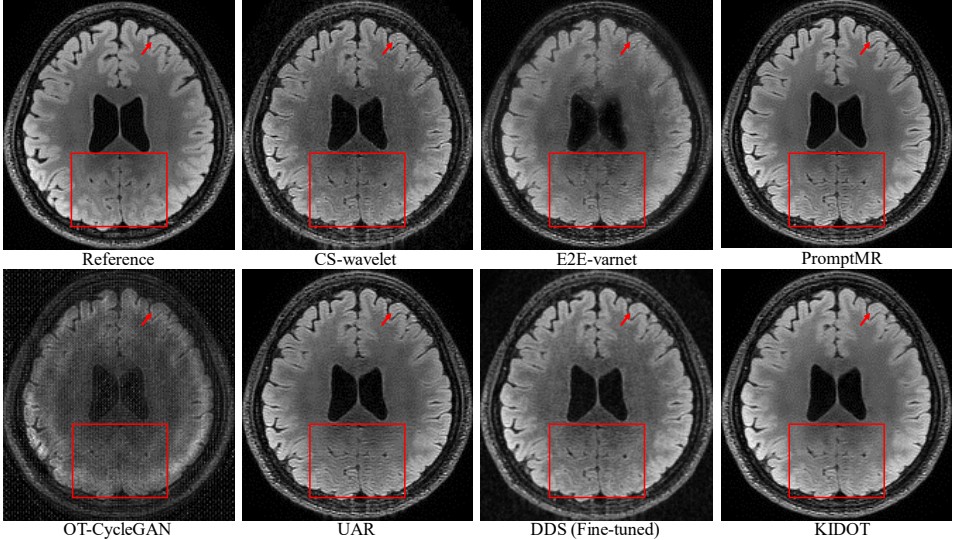

Figure 3: Qualitative comparison of MR reconstruction methods on the prospectively acquired United Imaging brain dataset (T2-FLAIR, 10x acceleration). KIDOT demonstrates enhanced detail recovery (e.g., red box).

**Clinical prospective CT data.** We finally evaluated KIDOT on a challenging clinical low-dose CT (LDCT) task, utilizing a dataset of prospectively acquired abdominal scans from 30 consented patients under Institutional Review Board approval. The LDCT protocol involved an approximate 10-fold reduction in radiation dose compared to the corresponding normal-dose CT (NDCT) scans. Crucially, differences in scan parameters and acquisition directions between the LDCT and NDCT protocols

resulted in significant spatial misalignments between the image pairs, a common complication in prospective clinical studies. From 28,251 available slices, 2,420 were selected for this evaluation. To benchmark KIDOT against methods reliant on aligned data, we generated simulated low-dose CT (SLDCT) images from the NDCT scans, injecting compound Gaussian and Poisson noise into the uncorrupted projections by using the method in [48, 49]. The transport network $\mathcal{T}_\phi$ within KIDOT for this task was based on the ISTA-Net architecture [50].

Due to the substantial spatial misalignments inherent in the dataset, performance was also primarily evaluated using the distribution-based metrics FID and KID. Table 3 presents the quantitative comparison. KIDOT achieves the best scores among all evaluated methods, outperforming conventional approaches like BM3D, supervised deep learning models such as REDCNN and the baseline ISTA-Net, and unpaired methods including OT-CycleGAN and UAR. This demonstrates KIDOT's strong capability to learn effective mappings for dynamic image reconstruction even from severely misaligned input distributions, reflecting its potential utility in practical clinical scenarios. Figure 4 provides qualitative results of LDCT data processed and the reconstruction achieved by various methods, including KIDOT,

Table 3: Quantitative comparison on **Clinical Prospective CT Data** (10-fold reduction). FID/KID are reported (lower is better ↓). Best results are highlighted in **bold**.

| Method | FID↓ | KID×100 ↓ |
|---|---|---|
| BM3D [51] | 100.82 | 6.12 |
| REDCNN [52] | 46.43 | 1.93 |
| ISTA-Net [50] | 24.86 | 1.59 |
| OT-CycleGAN [35] | 63.54 | 3.28 |
| UAR [37] | 23.82 | 1.45 |
| FSBM [41] | 68.42 | 3.61 |
| **KIDOT (ours)** | **22.25** | **1.32** |

underscoring the challenges posed by noise and the need for robust handling of real-world data complexities like misalignment. Among these, KIDOT's reconstruction (bottom-right) particularly stands out, demonstrating superior noise suppression and enhanced preservation of fine anatomical details compared to other approaches.

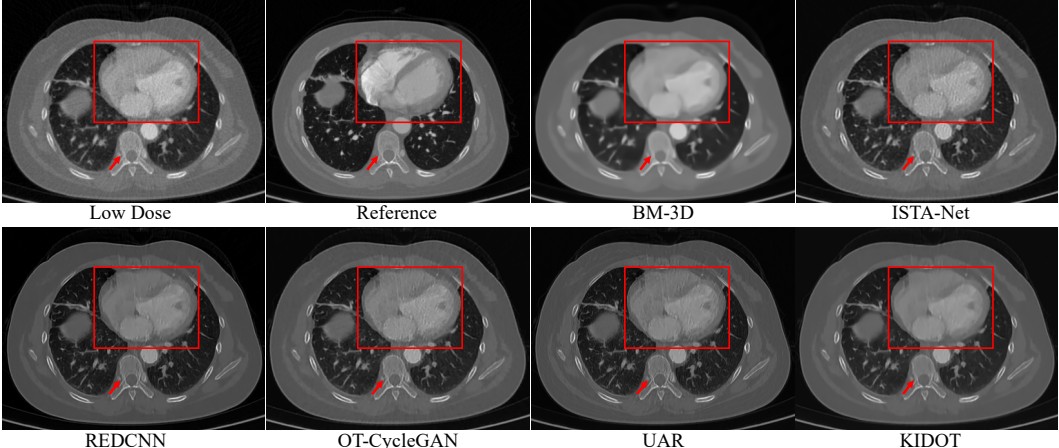

Figure 4: Qualitative results for LDCT reconstruction on clinical prospective data. Comparison illustrates noise reduction and detail enhancement achieved by KIDOT.

**Ablation Study.** We conduct an ablation study on the fastMRI knee dataset (4x acceleration) to evaluate the effectiveness of each component in KIDOT. We compared four different configurations: (1) training with only supervised loss $\mathcal{L}_{\text{SUP}}$ in Eq. (11), (2) training with the loss associated with $\varphi_\theta$ in Eq. (10) and $\mathcal{L}_{\text{SUP}}$, denoted as $\mathcal{L}_{\text{SUP}} + \varphi_\theta$, (3) adding the transport cost $C_{\text{final}} = \|y - \mathcal{A}(\mathcal{T}_\phi^{t_{N-1}}(y))\|_1$ for the final output in Eq. (10) to $\mathcal{L}_{\text{SUP}} + \varphi_\theta$, denoted as $\mathcal{L}_{\text{SUP}} + \varphi_\theta + C_{\text{final}}$, (4) using whole loss (KIDOT). Table 4 (upper two rows)

Table 4: Ablation study on the fastMRI knee dataset (4x acceleration).

| Method | PSNR | SSIM |
|---|---|---|
| $\mathcal{L}_{\text{SUP}}$ | 32.56 | 0.8372 |
| $\mathcal{L}_{\text{SUP}} + \varphi_\theta$ | 32.92 | 0.8422 |
| $\mathcal{L}_{\text{SUP}} + \varphi_\theta + C_{\text{final}}$ | 33.01 | 0.8425 |
| **KIDOT** | **33.31** | **0.8518** |

shows that adding the potential network $\varphi_\theta$ to the supervised baseline resulted in performance improvements. These gains indicate that $\varphi_\theta$ successfully guides the learning process towards better alignment with the target data distribution. It can be observed from Table 4 that incorporating the

imaging-knowledge-informed transport cost to the final output can further improve performance. Furthermore, our full KIDOT model that enforces consistency to imaging knowledge throughout the transport process, further improves the results, demonstrating the effectiveness of our imaging knowledge-informed dynamic OT for medical image reconstruction. The key difference between (3) and (4) lies in when and how the imaging- knowledge-informed cost is applied. The third setting represents an ablation where the physics-consistency cost is applied only to the final output of the N-step transport path. In contrast, our full KIDOT model enforces this physics consistency throughout the entire dynamic transport process, integrating the cost over all intermediate steps of the trajectory.

## 5 Conclusion and Limitations

We presented KIDOT, a novel framework for prospective medical image reconstruction. Our key innovation is framing reconstruction as a dynamic transport process whose learned path is explicitly constrained by imaging physics. Theoretical analysis and extensive experiments show the rationale and effectiveness of KIDOT. While KIDOT demonstrates promising results, it relies on the availability of the physical forward model (i.e., $\mathcal{A}$), which may limit its application when the information of $\mathcal{A}$ is missing. Meanwhile, the computational cost associated with simulating the dynamic transport path via numerical ODE integration can be higher than static methods. We will investigate more techniques in our framework to tackle these limitations in the future.

## Acknowledgement

This work was supported by National Key R&D Program(2022YFA1004201), NSFC (12090021, 12125104, 12401671, 12426313, 12501708, 12501709, 623B2084), Postdoctoral Research Funding Project of Shaanxi Province(2024BSHTDZZ007), China National Postdoctotal Program for Innovative Talents (BX20240276), China Fundamental Research Funds for the Central Universities (xzy022025047), China Postdoctoral Science Foundation (2025M773058), and the Key Laboratory of Biomedical Imaging Science and System, Chinese Academy of Sciences.

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

# A  Proofs and Algorithm

## A.1  KIDOT Training Algorithm

---

**Algorithm 1** KIDOT Training Algorithm

---

**Input:** Real prospective degraded dataset $\mathbb{Y}$ (samples $y \sim \mathbb{Y}$); high-quality dataset $\mathbb{X}$ (samples $x \sim \mathbb{X}$); unfolding transport network $\mathcal{T}_\phi$ and potential network $\varphi_\theta$; learning rates $\alpha_\phi, \alpha_\theta$; number of critic updates per generator update $N_c$.

 1: **while** $\phi$ has not converged **do**
 2:    **for** $k = 1, \cdots, N_c$ **do**
 3:       *% Training potential network $\varphi_\theta$.*
 4:       Sample mini-batch $\{y^{(j)}\}_{j=1}^B$ from $\mathbb{Y}$ and $\{x^{(j)}\}_{j=1}^B$ from $\mathbb{X}$.
 5:       Evolve $\{y^{(j)}\}$ to $\{\mathcal{T}_\phi^{t_{N-1}}(y^{(j)})\}$ using discrete transport equation for $N$ steps.
 6:       Compute critic loss: $\mathcal{L}_\theta \leftarrow \frac{1}{B}\sum_{j=1}^B \varphi_\theta(x^{(j)}) - \frac{1}{B}\sum_{j=1}^B \varphi_\theta(\mathcal{T}_\phi^{t_N}(y^{(j)}))$.
 7:       Update critic parameters: $\theta \leftarrow \theta + \alpha_\theta \nabla_\theta \mathcal{L}_\theta$.
 8:    **end for**
 9:    *% Training unfolding transport network $\mathcal{T}_\phi$.*
10:    Sample mini-batch $\{y^{(j)}\}_{j=1}^B$ from $\mathbb{Y}$.
11:    Evolve $\{y^{(j)}\}$ along paths $\{\mathcal{T}_\phi^{t_i}(y^{(j)})\}_{i=0}^{N-2}$ and to endpoint $\{\mathcal{T}_\phi^{t_{N-1}}(y^{(j)})\}$.
12:    Compute generator loss:
13:    $\mathcal{L}_\phi \leftarrow \frac{1}{B}\sum_{j=1}^B \left( \sum_{i=0}^{N-1} \frac{1}{N} \cdot \|y^{(j)} - \mathcal{A}(\mathcal{T}_\phi^{t_i}(y^{(j)}))\|_1 - \lambda\varphi_\theta(\mathcal{T}_\phi^{t_{N-1}}(y^{(j)})) \right)$.
14:    **if** paired data is used **then**
15:       Sample mini-batch $\{(y_p^{(j)}, x_p^{(j)})\}_{j=1}^{B_p}$ from $P_{pair}$.
16:       $\mathcal{L}_\phi \leftarrow \mathcal{L}_\phi + \frac{\gamma}{B_p}\sum_{j=1}^{B_p} \mathcal{F}(\mathcal{T}_\phi^{t_{N-1}}(y_p^{(j)}), x_p^{(j)})$.
17:    **end if**
18:    Update generator parameters: $\phi \leftarrow \phi - \alpha_\phi \nabla_\phi \mathcal{L}_\phi$.
19: **end while**

---

## A.2  Proof of Theorem 3.1

**Theorem A.1** (Existence and Geometry of Minimizer for $L^2$ Distance Integral)**.** *Let $x, y \in \mathbb{R}^n$ be distinct points and let $M \geq \|x - y\|_2$ be a constant. Define the feasible set of paths $\mathcal{X}_M$ as*

$$\mathcal{X}_M = \{s \in AC([0,1], \mathbb{R}^n) \mid s(0) = y, s(1) = x, \|s'\|_\infty \leq M\},$$

*where $AC([0,1], \mathbb{R}^n)$ denotes the space of absolutely continuous paths. Consider the optimization problem*

$$\inf_{s \in \mathcal{X}_M} I[s], \quad where \quad I[s] = \int_0^1 \|s(t) - y\|_2^2 \mathrm{d}t. \tag{13}$$

*Then, an optimal solution $s^* \in \mathcal{X}_M$ to problem (13) exists, and its geometric trajectory $\{s^*(t) \mid t \in [0,1]\}$ coincides with the straight line segment connecting $y$ and $x$.*

**Proof:** The proof proceeds in two parts. First, we establish the existence of an optimal solution. Second, we prove that the geometric trajectory of any such optimal solution must be the straight line segment connecting $y$ and $x$.

**Part 1: Existence of Optimal Solution**

We equip the space of continuous functions $C([0,1], \mathbb{R}^n)$ with the supremum norm $\|\cdot\|_\infty$, where $\|f\|_\infty = \sup_{t \in [0,1]} \|f(t)\|_2$. We next show that the feasible set $\mathcal{X}_M$ is nonempty and compact in this space and that the functional $I[s]$ is continuous, thus obtaining the existence of an optimal solution.

**1) Non-emptiness of $\mathcal{X}_M$.** Consider the straight line path $s_0(t) = (1 - t)y + tx$. It is absolutely continuous, $s_0(0) = y$, and $s_0(1) = x$. Its derivative is $s_0'(t) = x - y$ a.e., with $\|s_0'(t)\|_2 = \|x - y\|_2$. Since $M \geq \|x - y\|_2$, we have $\|s_0'\|_\infty = \|x - y\|_2 \leq M$. Thus, $s_0 \in \mathcal{X}_M$, and the feasible set is nonempty.

**2) Compactness of $\mathcal{X}_M$ (in $C([0,1], \mathbb{R}^n)$).** We apply the Arzelà-Ascoli theorem by verifying that $\mathcal{X}_M$ is closed, uniformly bounded, and equicontinuous.

- *Uniform Lipschitz Bound:* The condition $\|s'\|_\infty \leq M$ implies that every path $s \in \mathcal{X}_M$ is Lipschitz continuous with a uniform Lipschitz constant $M$, i.e., $\|s(t_1) - s(t_2)\|_2 \leq M|t_1 - t_2|$ for all $t_1, t_2 \in [0,1]$.

- *Closedness:* Let $\{s_n\}_{n=1}^\infty \subset \mathcal{X}_M$ be a sequence converging uniformly to $s \in C([0,1], \mathbb{R}^n)$, i.e., $\|s_n - s\|_\infty \to 0$. The limit function $s$ is continuous. Pointwise convergence implies $s(0) = \lim_{n\to\infty} s_n(0) = y$ and $s(1) = \lim_{n\to\infty} s_n(1) = x$. Taking the limit in the Lipschitz condition $\|s_n(t_1) - s_n(t_2)\|_2 \leq M|t_1 - t_2|$ yields $\|s(t_1) - s(t_2)\|_2 \leq M|t_1 - t_2|$ for all $t_1, t_2 \in [0,1]$. This shows $s$ is M-Lipschitz continuous. By Rademacher's theorem, a Lipschitz continuous function is differentiable almost everywhere, and at points where it is differentiable, its derivative $s'(t)$ satisfies $\|s'(t)\|_2 \leq M$. This implies that $s$ is absolutely continuous and its supremum norm satisfies $\|s'\|_\infty \leq M$. Thus, $s \in \mathcal{X}_M$, proving that $\mathcal{X}_M$ is closed in $C([0,1], \mathbb{R}^n)$.

- *Uniform Boundedness:* For any $s \in \mathcal{X}_M$ and $t \in [0,1]$, $\|s(t)\|_2 \leq \|s(0)\|_2 + \|s(t) - s(0)\|_2 \leq \|y\|_2 + M|t - 0| \leq \|y\|_2 + M$. Hence, $\|s\|_{L^\infty} \leq \|y\|_2 + M$ for all $s \in \mathcal{X}_M$, making the set uniformly bounded.

- *Equicontinuity:* Since all functions in $\mathcal{X}_M$ share the same Lipschitz constant $M$, the family $\mathcal{X}_M$ is equicontinuous. Specifically, for any $\epsilon > 0$, choose $\delta = \epsilon/M$ ($M > 0$). Then for all $s \in \mathcal{X}_M$, $|t_1 - t_2| < \delta$ implies $\|s(t_1) - s(t_2)\|_2 \leq M|t_1 - t_2| < M\delta = \epsilon$.

By the Arzelà-Ascoli theorem, $\mathcal{X}_M$ is compact in $C([0,1], \mathbb{R}^n)$.

**3) Continuity of the Functional $I[s]$.** Let $s_n \to s$ uniformly in $C([0,1], \mathbb{R}^n)$. Then

$$|I[s_n] - I[s]| \leq \int_0^1 |\|s_n(t) - y\|_2^2 - \|s(t) - y\|_2^2|\mathrm{d}t$$

$$= \int_0^1 |\langle s_n(t) - s(t), s_n(t) + s(t) - 2y\rangle|\mathrm{d}t$$

$$\leq \int_0^1 \|s_n(t) - s(t)\|_2\|s_n(t) + s(t) - 2y\|_2\mathrm{d}t.$$

Since $\{s_n\}$ is uniformly bounded, $\|s_n(t) + s(t) - 2y\|_2$ is bounded by some constant $C'$. Therefore,

$$|I[s_n] - I[s]| \leq C' \int_0^1 \|s_n(t) - s(t)\|_2\mathrm{d}t \leq C' \int_0^1 \|s_n - s\|_\infty\mathrm{d}t = C'\|s_n - s\|_\infty.$$

As $\|s_n - s\|_\infty \to 0$, we have $|I[s_n] - I[s]| \to 0$. Hence, $I[s]$ is continuous on $C([0,1], \mathbb{R}^n)$ with the supremum norm.

**4) Existence Conclusion.** By the Weierstrass Extreme Value Theorem, a continuous real-valued functional ($I[s]$) defined on a non-empty compact set ($\mathcal{X}_M$) attains its minimum value on that set. Therefore, there exists at least one optimal solution $s^* \in \mathcal{X}_M$ for problem (13).

**Part 2: Geometry of the Optimal Trajectory**

Let $s^* \in \mathcal{X}_M$ be an optimal solution, which exists by Part 1. We show that its geometric trajectory must be the straight line segment connecting $y$ and $x$. Let $I$ denote this line segment:

$$I = \{z \in \mathbb{R}^n \mid \exists s \in [0,1], \, z = (1-s)y + sx\}.$$

For any $t \in [0,1]$, let $q(t) = \mathrm{proj}_I(s^*(t))$ be the orthogonal projection of $s^*(t)$ onto $I$.

Assume, for the sake of contradiction, that the geometric trajectory of $s^*$ is not the line segment $I$. This implies that the set $T = \{t \in [0,1] \mid s^*(t) \notin I\}$ has a positive Lebesgue measure. For $t \in T$, $s^*(t) \neq q(t)$. By the property of orthogonal projection (Pythagorean theorem), for any $t \in [0,1]$:

$$\|s^*(t) - y\|_2^2 = \|s^*(t) - q(t)\|_2^2 + \|q(t) - y\|_2^2.$$

Since $\|s^*(t) - q(t)\|_2^2 \geq 0$, we have $\|s^*(t) - y\|_2^2 \geq \|q(t) - y\|_2^2$ for all $t \in [0,1]$. Crucially, for $t \in T$, since $s^*(t) \neq q(t)$, we have $\|s^*(t) - q(t)\|_2^2 > 0$, which implies the strict inequality:

$$\|s^*(t) - y\|_2^2 > \|q(t) - y\|_2^2, \quad \forall t \in T.$$

Now, we integrate over $[0, 1]$ obtaining

$$I[s^*] = \int_0^1 \|s^*(t) - y\|_2^2 \mathrm{d}t = \int_T \|s^*(t) - y\|_2^2 \mathrm{d}t + \int_{[0,1]\setminus T} \|s^*(t) - y\|_2^2 \mathrm{d}t$$

$$> \int_T \|q(t) - y\|_2^2 \mathrm{d}t + \int_{[0,1]\setminus T} \|q(t) - y\|_2^2 \mathrm{d}t$$

$$= \int_0^1 \|q(t) - y\|_2^2 \mathrm{d}t = I[q].$$

where the inequality is because $T$ has a positive Lebesgue measure. Thus, we have $I[s^*] > I[q]$. Next, we verify that the projected path $q(t)$ is feasible, i.e., $q \in \mathcal{X}_M$.

- *Boundary Conditions:* Since $s^*(0) = y \in I$ and $s^*(1) = x \in I$, their projections are themselves: $q(0) = \text{proj}_I(y) = y$ and $q(1) = \text{proj}_I(x) = x$.

- *Absolute Continuity and Speed Bound:* The orthogonal projection onto a convex set (like the line segment $I$ or the line containing it) is 1-Lipschitz (non-expansive), i.e., $\|\text{proj}_I(a) - \text{proj}_I(b)\|_2 \leq \|a - b\|_2$. Therefore,

$$\|q(t_1) - q(t_2)\|_2 = \|\text{proj}_I(s^*(t_1)) - \text{proj}_I(s^*(t_2))\|_2 \leq \|s^*(t_1) - s^*(t_2)\|_2.$$

Since $s^* \in \mathcal{X}_M$ is M-Lipschitz, $q(t)$ is also M-Lipschitz. This implies $q(t)$ is absolutely continuous and almost everywhere differentiable, its derivative satisfies $\|q'(t)\|_2 \leq M$ almost everywhere, hence, $\|q'\|_\infty \leq M$.

Thus, $q(t)$ belongs to the feasible set $\mathcal{X}_M$. We have now found a feasible path $q \in \mathcal{X}_M$ such that $I[q] < I[s^*]$. This contradicts the assumption that $s^*$ is an optimal solution (minimizer) for problem (13). Therefore, the geometric trajectory of any optimal solution $s^*$ must be the straight line segment connecting $y$ and $x$.

$\square$

## A.3 Proof of Theorem A.2

We make the assumptions: A1) Measures $\mathbb{P}$ and $\mathbb{Q}$ have compact support, and the parameter set $K \subset \mathbb{R}^d$ is non-empty and compact. A2) $\mathcal{T}_{\phi_n}^t \to \mathcal{T}_\phi^t$ pointwise whenever $\phi_n \to \phi$ and there exists $h \in L^1([0,1])$ such that $|c(\mathcal{T}_{\phi_n}^t(y), y)| \leq h(t)$ for all $n$ and $t \in [0,1]$. A3) $\sup_{\phi \in K} \|\mathcal{T}_\phi^t\|_\infty < \infty$. A4) For fixed $y$, the cost function $c(y, \cdot)$ is continuous in its second argument.

**Theorem A.2** (Existence of Minimizer for KIDOT Objective)**.** *Suppose the assumptions A1-A4 hold, then there exists a minimizer for the optimization problem*

$$\inf_{\phi \in K} J(\phi | \lambda, \mathbb{P}, \mathbb{Q}) := \mathbb{E}_{I_0 \sim \mathbb{P}} \left[ \int_0^1 c(\mathcal{T}_\phi^t(I_0), I_0) \, \mathrm{d}t \right] + \lambda W_1(\mathcal{T}_{\phi\#}^1(\mathbb{P}), \mathbb{Q}). \tag{14}$$

**Proof:** Let $\{\mathcal{T}_{\phi_n}^t\}$ be a minimizing sequence such that

$$\lim_{n\to\infty} J(\mathcal{T}_{\phi_n}^t \mid \lambda, \mathbb{P}) = \inf_\phi J(\mathcal{T}_\phi^t \mid \lambda, \mathbb{P}). \tag{15}$$

Since $\phi_n \in K$, where $K$ is compact and finite-dimensional, there exists a convergent subsequence, up to sub-sequences, (still indexed by $n$) such that $\phi_n \to \phi^*$ for some $\phi^* \in K$. By assumption, this implies pointwise convergence:

$$\mathcal{T}_{\phi_n}^t(y) \to \mathcal{T}_{\phi^*}^t(y) \quad \text{for all } y \in \mathbb{P}, \ t \in [0,1].$$

We next show that $\mathcal{T}_{\phi^*}^t$ is a minimizer of (14). By the continuity of $c(y, \cdot)$ in its second argument, we have

$$\int_0^1 c\big(y, \mathcal{T}_{\phi_n}^t(y)\big) \mathrm{d}t \to \int_0^1 c\big(y, \mathcal{T}_{\phi^*}^t(y)\big) \mathrm{d}t.$$

Moreover, since $\left|c\big(y, \mathcal{T}^t_{\phi_n}(y)\big)\right|$ is uniformly bounded by an integrable function $h(t)$, the Dominated Convergence Theorem yields

$$\lim_{n\to\infty} \mathbb{E}_{\mathbb{P}}\Big[\int_0^1 c\big(y, \mathcal{T}^t_{\phi_n}(y)\big)\mathrm{d}t\Big] = \mathbb{E}_{\mathbb{P}}\Big[\int_0^1 c\big(y, \mathcal{T}^t_{\phi^*}(y)\big)\mathrm{d}t\Big]. \tag{16}$$

Next, for any $\varphi \in C_b(\mathbb{R}^k)$, we observe

$$\left|\int \varphi(x)\, d(\mathcal{T}^t_{\phi_n})_\#\mathbb{P} - \int \varphi(x)\, d(\mathcal{T}^t_{\phi^*})_\#\mathbb{P}\right| \le \int \left|\varphi(\mathcal{T}^t_{\phi_n}(y)) - \varphi(\mathcal{T}^t_{\phi^*}(y))\right| d\mathbb{P}(y) \to 0,$$

which implies that $(\mathcal{T}^t_{\phi_n})_\#\mathbb{P}$ converges narrowly to $(\mathcal{T}^t_{\phi^*})_\#\mathbb{P}$ as $n \to \infty$.

Additionally, since $\sup_{\phi \in K} \|\mathcal{T}^t_\phi\|_\infty < \infty$, we apply Dominated Convergence Theorem once again to obtain

$$\left|\int \|x\|_2\, d(\mathcal{T}^t_{\phi_n})_\#\mathbb{P} - \int \|x\|_2\, d(\mathcal{T}^t_{\phi^*})_\#\mathbb{P}\right| \to 0.$$

By [53, Theorem 5.11], narrow convergence and convergence of first moments together imply that

$$W_1\big((\mathcal{T}^t_{\phi_n})_\#\mathbb{P}, (\mathcal{T}^t_{\phi^*})_\#\mathbb{P}\big) \to 0.$$

By the triangle inequality, it follows that

$$\lim_{n\to\infty} W_1\big(\mathbb{Q}, (\mathcal{T}^t_{\phi_n})_\#\mathbb{P}\big) = W_1\big(\mathbb{Q}, (\mathcal{T}^t_{\phi^*})_\#\mathbb{P}\big). \tag{17}$$

Combining Eq. (15), (16), (17), we obtain

$$\begin{aligned}
J(\mathcal{T}^t_{\phi^*} \mid \lambda, \mathbb{P}) &= \mathbb{E}_{\mathbb{P}}\Big[\int_0^1 c\big(y, \mathcal{T}^t_{\phi^*}(y)\big)\mathrm{d}t\Big] + \lambda W_1\big(\mathbb{Q}, (\mathcal{T}^t_{\phi^*})_\#\mathbb{P}\big) \\
&\overset{(16)(17)}{=} \lim_{n\to\infty} \mathbb{E}_{\mathbb{P}}\Big[\int_0^1 c\big(y, \mathcal{T}^t_{\phi_n}(y)\big)\mathrm{d}t\Big] + \lambda W_1\big(\mathbb{Q}, (\mathcal{T}^t_{\phi_n})_\#\mathbb{P}\big) \\
&= \lim_{n\to\infty} J(\mathcal{T}^t_{\phi_n} \mid \lambda, \mathbb{P}) \\
&\overset{(15)}{=} \inf_\phi J(\mathcal{T}^t_\phi \mid \lambda, \mathbb{P}).
\end{aligned}$$

This completes the proof that $\mathcal{T}^t_{\phi^*}$ is a minimizer of (14).

$\square$

# B Additional Experiments

## B.1 Implementation Details and Baselines

This section introduces the compared methods and detailed settings in our experiments.

**Implementation details.** We trained distinct KIDOT models tailored to each specific datasets (simulated MRI data, real prospective MRI data and clinical prospective CT data). Optimization utilized the RMSProp algorithm with differential learning rates: $1 \times 10^{-4}$ for the transport network ($\mathcal{T}_\phi$) and $2 \times 10^{-4}$ for the potential network ($\varphi_\theta$). The inner iteration parameter $N_c$ was consistently set to 1. A learning rate decay schedule applied a factor of 10 reduction after each block of 30 training epochs. The underlying architecture for $\mathcal{T}_\phi$ was adapted based on the dataset: using the E2E-VarNet backbone [12] for simulated MRI data, the PromptMR backbone [13] for real prospective MRI data, and the MC-CDic backbone [50] for clinical prospective CT data. Experiments were performed using PyTorch on an NVIDIA 4090 GPU. The source code will be publicly released following possible publication.

**Representative compared methods.** In MRI reconstruction experiments, we conducted comparisons against several existing methods, including the traditional CS-wavelet [45], two representative deep learning unfolding networks, E2E-VarNet [12] and PromptMR [13], a diffusion-based method (DDS [47]) and three comparative methods based on OT: OT-CycleGAN [35], UAR [37] and FSBM [41].

For CT reconstruction, our set of comparison methods consists of the traditional BM3D [51], two representative deep learning methods (ISTA-Net [50], an unfolding network, and REDCNN [52]), and three OT-based approaches, OT-CycleGAN [35], UAR [37] and FSBM [41].

**Motivation for Transport Network Implementation** Our motivation for the implementation of the transport network, $\mathcal{T}_\phi$, is guided by three core principles: leveraging task-specific knowledge, ensuring fair comparisons, and demonstrating the broad applicability of our KIDOT framework.

First, we utilize backbones incorporated with task-specific knowledge. In medical image reconstruction, different imaging modalities, such as MRI and CT, possess unique physical properties and data characteristics. For example, MRI heavily relies on multi-coil acquisition, where effectively leveraging the physical information from coil sensitivity maps is often critical for optimal reconstruction. Meanwhile, the imaging mechanism in CT and MRI are different. CT images are produced by Radon Transform and MR images are produced by Fourier Transform. This necessity to incorporate modality-specific knowledge has led the research community to develop specialized, state-of-the-art architectures for each task. To evaluating the performance of our approach for each task, we adopt the state-of-the-art architectures for each task. Note that, for MRI, E2E-VarNet is more light-weight than PromptMR, and we use E2E-VarNet for simulated MRI for the sake of efficient computation.

Second, we ensure a fair comparison in each experiment. For any given task, the transport network in our KIDOT framework uses the same backbone architecture as the corresponding supervised baseline it is compared against. For example, when comparing with E2E-VarNet on the simulated MRI dataset, our KIDOT model also uses an E2E-VarNet-based architecture for its transport network $\mathcal{T}_\phi$.

Finally, this approach allows us to demonstrate the versatile usage of KIDOT on different backbones. By deliberately choosing backbones that are identical or highly similar to the supervised baselines, we effectively isolate the contribution of our framework. This ensures that the observed performance gain is attributable to our novel knowledge-informed dynamic optimal transport formulation itself, rather than simply employing a more powerful (and different) network architecture.

## B.2 Datasets Details

**Simulated MRI data.** All images were center-cropped to 320×320 pixels. We simulated undersampled measurements by applying k-space masks equivalent to a 4× acceleration factor. A key aspect of this evaluation was to assess KIDOT's robustness to scenarios where test data characteristics differ from training data, a common challenge in prospective clinical deployment. To achieve this, we generated two distinct sets of undersampled inputs: one set, with specific sampling patterns, was used for training supervised baselines, while a different set of sampling patterns was employed for testing to mimic prospective acquisition conditions. Specifically, the test mask was generated by randomly flipping 3% of the entries in the training mask. Examples of these distinct masks are shown in Figure. 5.

**Real prospective MRI data.** For real prospective MRI data experiments, we used a multi-contrast MRI dataset acquired on a 3.0T United Imaging scanner equipped with a 32-channel receiver head coil. We selected the T2-FLAIR weighted sequence from this dataset. The T2-FLAIR images were acquired using a 2D Inversion Recovery (IR) sequence with the following parameters: TR=6000 ms, TE=396.44 ms, and volume size 256×256×176. The dataset was partitioned into training, validation, and test sets consisting of 3077, 1860, and 3077 slices, respectively.

**Clinical prospective CT data.** For our clinical prospective CT reconstruction evaluation, we utilized a dataset of real Low-Dose CT (LDCT) and Normal-Dose CT (NDCT) images. Data acquisition was approved by the Institutional Review Board of the xxx hospital. Each subject underwent triple-phase CT scans—arterial, portal venous, and delayed—on a United Imaging Healthcare (UIH) uCT960+ 320-slice helical scanner (rotation time 0.5 s/rotation, pitch 0.9937, collimation 80 mm; reconstructed via UIH uInnovation-CT Explorer R001). Scans exhibited similar anatomical structures

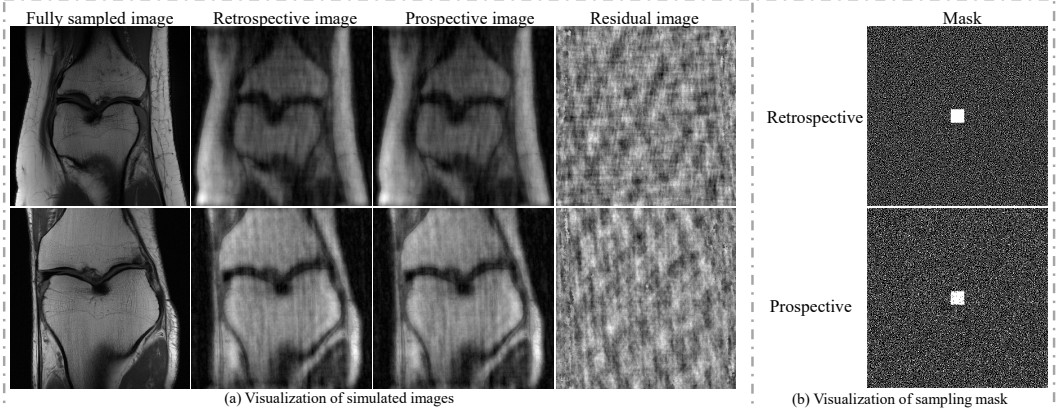

|  Fully sampled image | Retrospective image | Prospective image | Residual image | | Mask |

(a) Visualization of simulated images

(b) Visualization of sampling mask

Figure 5: (a) Visualization of simulated images: the first column shows fully sampled images, the second column is used for supervised learning, the third column represents the undersampled prospective data, and the fourth column shows the residuals (difference between supervised and prospective images). (b) Visualization of simulated images:the first row is the retrospective mask, and the second row is the prospective mask.

and consistent slice counts but differed in contrast enhancement. The dataset involved 30 consented patients with breathing training. Bolus tracking was used (abdominal aorta, 150 HU threshold). NDCT scans (head-to-foot, 100 kVp, 213 mAs DL 2) were triggered at 16s, 50s, and 120s post-threshold. LDCT scans (foot-to-head, 80 kVp, 47 mAs, 1/10th NDCT dose) followed immediately. Slice counts varied (98-645), yielding 28,251 512×512 abdominal LDCT/NDCT images. We used a subset of 2420 pairs.

Given the potential for anatomical misalignment in real prospective acquisitions (due to different scan directions/timings), we also generated perfectly-paired Simulated Low-Dose CT (SLDCT) images by adding noise in the projection domain. We inject compound Gaussian and Poisson noise into the uncorrupted projections by using the method in [48, 49]. The incident scale photon flux is 40000.We adopt a fan-beam geometry to simulate the uncorrupted projections from the CT images. The geometrical parameters for projection were: source-to-isocenter distance 570.0 mm, source-to-detector distance 962.9 mm, image size $512 \times 512$ (with pixel spacing $0.6934 \times 0.6934$ mm$^2$), detector bin number 864, detector bin width 1.0336 mm, and 1200 projection views acquired over a 360-degree orbit.

To visualize the characteristics of these different CT data types and the discrepancy between our simulated low-dose data and real clinical low-dose data, we provide examples in Figure 6. From left to right, the figure shows: the normal dose image, the generated simulated low dose image, a corresponding real low dose image from our prospective dataset, and the residual image calculated as the difference between the SLDCT and LDCT images (SLDCT - LDCT). This residual highlights aspects of real low-dose noise and artifacts not fully captured by our simulation model.

## B.3 Additional Ablation Studies

### B.3.1 Effect of Different Number of Discretization Steps $N$

We conduct an ablation study to investigate the sensitivity of our method to the number of discretization steps $N$, which is a key hyperparameter in our framework. The choice of $N$ represents a critical trade-off between the fidelity of the ODE approximation and the associated computational cost. A smaller $N$ can lead to a coarse and inaccurate discretization of the continuous transport path, resulting in suboptimal reconstruction quality. Conversely, a large $N$ increases the computational burden, as inference time and memory grow linearly with $N$, and can also introduce training instability due to the risk of gradient exploding or vanishing in deeper backpropagation paths.

This analysis is performed on the real prospective MRI data with a $10\times$ acceleration factor to find the optimal balance. The results are reported in Table 5. As shown, the reconstruction quality, measured by both FID and KID, consistently improves as $N$ increases from 6 to 12, achieving the

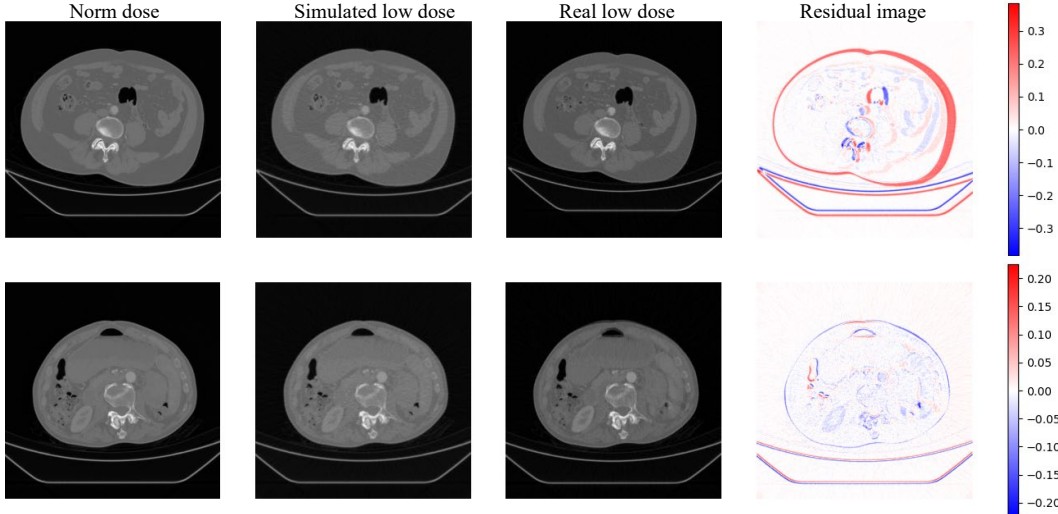

Figure 6: Comparison of Normal dose, Simulated low dose, and Real low dose CT Images with Residual images calculated as the difference between the SLDCT and LDCT images.

optimal results at $N = 12$. However, when we further increase the number of steps to $N = 15$, the performance begins to degrade. This degradation is likely attributable to the increased possibility of gradient exploding/vanishing during training. Based on this analysis, we fix $N = 12$ for our experiments.

Table 5: Comparison of different number of discretization steps $N$ on the **Real Prospective MRI Data** ($10\times$ acceleration). Best results are **bolded**.

| discretization steps $N$ | FID $\downarrow$ | KID$\times100 \downarrow$ |
|:---:|:---:|:---:|
| 6 | 35.31 | 2.01 |
| 9 | 30.06 | 1.62 |
| 12 | **28.26** | **1.12** |
| 15 | 29.23 | 1.31 |

### B.3.2 Sensitivity to the Supervised Trade-off Parameter $\gamma$

We analyze the sensitivity of KIDOT to the supervised trade-off parameter $\gamma$ using simulated MRI data ($4\times$ acceleration). As shown in Table 6, the choice of $\gamma$ impacts KIDOT's performance metrics. We observe a general trend where increasing $\gamma$ from 1 leads to consistent improvements in reconstruction quality (higher PSNR and SSIM) and fidelity (lower FID and KID). The best results across all evaluated metrics are achieved when $\gamma = 10^4$. Further increasing the trade-off parameter to $\gamma = 10^5$ results in a slight degradation across all metrics compared to $\gamma = 10^4$. Based on this result, we fix $\gamma = 10^4$ in our experiments.

Table 6: Comparison of different trade-off parameter $\gamma$ on the **Simulated MRI data** ($4\times$ acceleration). Best results are **bolded**.

| trade-off parameter $\gamma$ | PSNR $\uparrow$ | SSIM $\uparrow$ | FID $\downarrow$ | KID$\times100 \downarrow$ |
|:---:|:---:|:---:|:---:|:---:|
| 1 | 30.87 | 0.7815 | 54.82 | 2.47 |
| 10 | 31.71 | 0.7987 | 37.74 | 1.52 |
| $10^2$ | 32.28 | 0.8204 | 29.52 | 1.38 |
| $10^3$ | 33.04 | 0.8408 | 17.23 | 1.01 |
| $10^4$ | **33.31** | **0.8518** | **10.68** | **0.83** |
| $10^5$ | 33.10 | 0.8490 | 15.39 | 0.97 |

**B.3.3   Effect of Different Lagrange Multiplier $\lambda$**

We investigate the effect of the Lagrange multiplier $\lambda$ on KIDOT's performance using simulated MRI data ($4\times$ acceleration). Table 7 summarizes the results for different values of $\lambda$. We observe that the choice of $\lambda$ has an impact on both reconstruction quality and fidelity metrics. Notably, the optimal performance across all evaluated metrics (highest PSNR/SSIM and lowest FID/KID) is achieved with a Lagrange multiplier of $\lambda = 1$. Increasing $\lambda$ beyond this value leads to a consistent and substantial degradation in performance. As $\lambda$ increases to $10^2$, PSNR and SSIM decrease, while FID and KID$\times 100$ values increase drastically, indicating a considerable loss of image fidelity and introduces more artifacts. Based on these results, we fix $\lambda = 1$ for our experiments.

Table 7: Comparison of different Lagrange multiplier $\lambda$ on the **Simulated MRI data.** ($4\times$ acceleration). Best results are **bolded**.

| Lagrange multiplier $\lambda$ | PSNR $\uparrow$ | SSIM $\uparrow$ | FID $\downarrow$ | KID$\times 100 \downarrow$ |
|---|---|---|---|---|
| 0.5 | 33.07 | 0.8469 | 18.43 | 1.01 |
| 1 | **33.31** | **0.8518** | **10.68** | **0.83** |
| 2 | 32.94 | 0.8457 | 20.14 | 1.07 |
| 5 | 32.87 | 0.8436 | 26.69 | 1.25 |
| 10 | 27.62 | 0.8132 | 113.43 | 6.21 |
| $10^2$ | 26.15 | 0.7518 | 178.59 | 7.33 |

**B.3.4   Effect of the Physical Forward Operator $\mathcal{A}$ in Transport Equation**

We conducted an ablation study to evaluate the importance of explicitly incorporating the physical forward operator $\mathcal{A}$ within our KIDOT framework. This comparison was performed on the simulated MRI data with a $4\times$ acceleration factor. The setting "w/o $\mathcal{A}$" indicates that the physical forward operator was removed from the transport equation, or the network architecture $\mathcal{T}_\phi$. As summarized in Table 8, including the physical forward operator $\mathcal{A}$ leads to a substantial improvement in reconstruction quality and fidelity. Comparing the setting "w $\mathcal{A}$" to "w/o $\mathcal{A}$", we observe that incorporating $\mathcal{A}$ results in a notable increase in PSNR (33.31 vs 30.88) and SSIM (0.8518 vs 0.8174). The fidelity metrics are improved, with FID decreasing from 36.53 to 10.68, and KID$\times 100$ decreasing from 1.48 to 0.83. These results clearly demonstrate the crucial role of the physical forward operator in guiding the reconstruction process and highlight its contribution to KIDOT's superior performance.

Table 8: Results of the method w/wo physical forward operator $\mathcal{A}$ on the **Simulated MRI data** ($4\times$ acceleration). Best results are **bolded**.

| Setting | PSNR $\uparrow$ | SSIM $\uparrow$ | FID $\downarrow$ | KID$\times 100 \downarrow$ |
|---|---|---|---|---|
| w $\mathcal{A}$ | **33.31** | **0.8518** | **10.68** | **0.83** |
| w/o $\mathcal{A}$ | 30.88 | 0.8174 | 36.53 | 1.48 |

**B.4   Statistical Significance Analysis**

We perform statistical significance tests to compare baseline methods against our proposed KIDOT method on the simulated MRI dataset with $4\times$ acceleration. Table 9 summarizes the p-values from these tests. Each test compares the results of a baseline method directly against KIDOT.

For image-level quality metrics (PSNR and SSIM), p-values are computed via paired t-tests on the per-image metric values. Specifically, for each image, we obtain the metric values from all runs for both methods, then conduct paired t-tests across the set of images, thus accounting for within-image variability and dependencies between paired samples.

For distribution-based fidelity metrics (FID and KID$\times 100$), which are traditionally computed once per dataset, we obtain multiple samples for statistical testing using a bootstrap resampling procedure over the test images within each run. The FID and KID scores computed on these bootstrap samples form the basis for independent two-sample t-tests comparing baseline methods with KIDOT.

As shown in Table 9, all p-values fall below the conventional significance threshold of 0.05. Notably, comparisons against CS-wavelet, E2E-VarNet, and OT-CycleGAN yield p-values $< 0.001$ across all metrics, indicating strong statistical significance. For UAR, p-values range from 0.009 (KID$\times$100) to 0.024 (PSNR), still demonstrating significant improvements by KIDOT. These results collectively confirm that KIDOT outperforms all tested baselines on the simulated MRI dataset with high statistical confidence.

Table 9: Statistical significance (p-values) of baseline methods compared to **KIDOT** on the Simulated MRI data ($4\times$ acceleration). Lower p-values indicate more significant differences. P-values are from independent t-tests.

| Method | PSNR | SSIM | FID | KID$\times$100 |
|---|---|---|---|---|
| CS-wavelet | $< 0.001$ | $< 0.001$ | $< 0.001$ | $< 0.001$ |
| E2E-VarNet | $< 0.001$ | $< 0.001$ | $< 0.001$ | $< 0.001$ |
| OT-CycleGAN | $< 0.001$ | $< 0.001$ | $< 0.001$ | $< 0.001$ |
| UAR | 0.024 | $< 0.001$ | $< 0.001$ | 0.009 |

## B.5  Training Cost Curves for Three Datasets

In Figure 7, we display the cost curves over three datasets (i.e., simulated MRI data, real prospective MRI data and clinical prospective CT data) of $\mathcal{T}_\phi$ and $\varphi_\theta$ in the training process. The $\mathcal{T}_\phi$ cost curve is normalized to [0, 1]. $\varphi_\theta$ cost is scaled to [0, 1] and then take the negative.

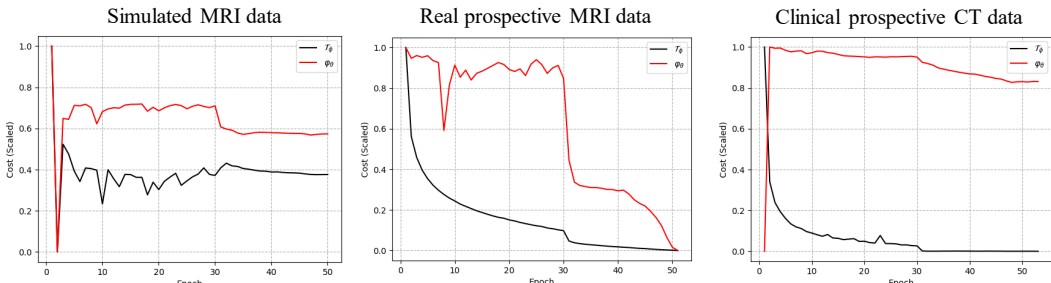

Figure 7: The training cost curves for two datasets. The cost of $\mathcal{T}_\phi$ is scaled to [0, 1]. The cost of $\varphi_\theta$ is scaled to [0, 1] and then takes the negative.

## B.6  Performance in Low-data Regimes

To evaluate the robustness of our method under data scarcity, we conduct an additional ablation study in a low-data regime. We intentionally restrict the amount of available training data by using reduced subsets of the simulated MRI dataset. Specifically, we train our model and the baselines on only 20% and 10% of the original paired data.

The results are presented in Table 10. In both settings, our KIDOT framework consistently outperforms the E2E-VarNet and UAR baselines across all metrics. This improved performance under data scarcity can be directly attributed to the regularization imposed by KIDOT's physics-informed constraints. Unlike purely data-driven methods that are prone to overfitting on small datasets, our framework's adherence to the physical model provides a robust inductive bias, significantly reducing its dependency on the volume of training data and enhancing its generalization capability.

Table 10: Performance comparison in low-data regimes on the **Simulated MRI data**. We train all methods using 20% and 10% of the original training paired data. Best results for each setting are **bolded**.

| Data | Method | PSNR ↑ | SSIM ↑ | FID ↓ | KID×100 ↓ |
|------|--------|--------|--------|-------|-----------|
| 20% | E2E-VarNet | 30.97 | 0.7978 | 72.43 | 4.02 |
| | UAR | 31.30 | 0.8054 | 40.32 | 2.95 |
| | KIDOT (Ours) | **31.75** | **0.8131** | **31.61** | **2.32** |
| 10% | E2E-VarNet | 29.92 | 0.7681 | 92.41 | 6.07 |
| | UAR | 30.64 | 0.7857 | 87.07 | 5.68 |
| | KIDOT (Ours) | **30.88** | **0.7952** | **73.14** | **4.11** |

## B.7 Generalization to More Aggressive Out-of-Distribution (OOD) or Heterogeneous Data

To further investigate the robustness and generalization capabilities of our model, we conducted two challenging cross-domain experiments on the fastMRI dataset, evaluating performance on both cross-anatomy and cross-contrast tasks. These experiments are designed to assess how well the models learn the fundamental principles of MRI reconstruction, rather than simply memorizing features specific to the training data distribution.

**Cross-Anatomy (Knee → Brain).** In this setup, the models are tasked with reconstructing brain images after being trained on knee data. For the supervised baseline (E2E-VarNet), we used paired knee scans for training. For our unpaired method, KIDOT, we trained it using unpaired data consisting of high-quality knee scans and low-quality brain scans. Both models were then evaluated on their ability to reconstruct brain scans. As shown in Table 11, KIDOT significantly outperforms E2E-VarNet, demonstrating its ability to generalize anatomical knowledge.

Table 11: Generalization performance on the Cross-Anatomy (Knee → Brain) task. Best results are **bolded**.

| Method | PSNR ↑ | SSIM ↑ | FID ↓ | KID×100 ↓ |
|--------|--------|--------|-------|-----------|
| E2E-VarNet | 33.82 | 0.7880 | 80.21 | 4.24 |
| **KIDOT (Ours)** | **34.62** | **0.8145** | **26.87** | **1.53** |

**Cross-Contrast (T1 → T2).** Similarly, this experiment tests generalization across different MRI contrasts. The models were trained on a dataset of unpaired high-quality T1-weighted brain scans and low-quality T2-weighted brain scans. The evaluation was then performed on the task of reconstructing T2-weighted brain scans. The results are presented in Table 12. Again, KIDOT shows superior performance, indicating its robustness to variations in image contrast.

Table 12: Generalization performance on the Cross-Contrast (T1 → T2) task. Best results are **bolded**.

| Method | PSNR ↑ | SSIM ↑ | FID ↓ | KID×100 ↓ |
|--------|--------|--------|-------|-----------|
| E2E-VarNet | 32.06 | 0.8229 | 31.29 | 1.42 |
| **KIDOT (Ours)** | **32.48** | **0.8348** | **29.74** | **1.33** |

The superior performance of KIDOT in both of these challenging OOD scenarios provides strong evidence for the benefit of its physics-guided inductive bias. This bias encourages the model to learn a generalizable reconstruction process grounded in the principles of MR imaging, rather than overfitting to the specific anatomical or contrast features of the training domain. While a standard supervised baseline performs well on in-domain data, it struggles when faced with a significant domain shift, as its learned prior is purely data-driven and less generalizable.

## B.8 Additional Visual Results

### B.8.1 Additional Visual Comparison on Simulated MRI Data

Figure 8 presents additional visual comparisons on simulated MRI data, highlighting that our method reconstructs clearer structural details with fewer artifacts in the regions of interest (red boxes).

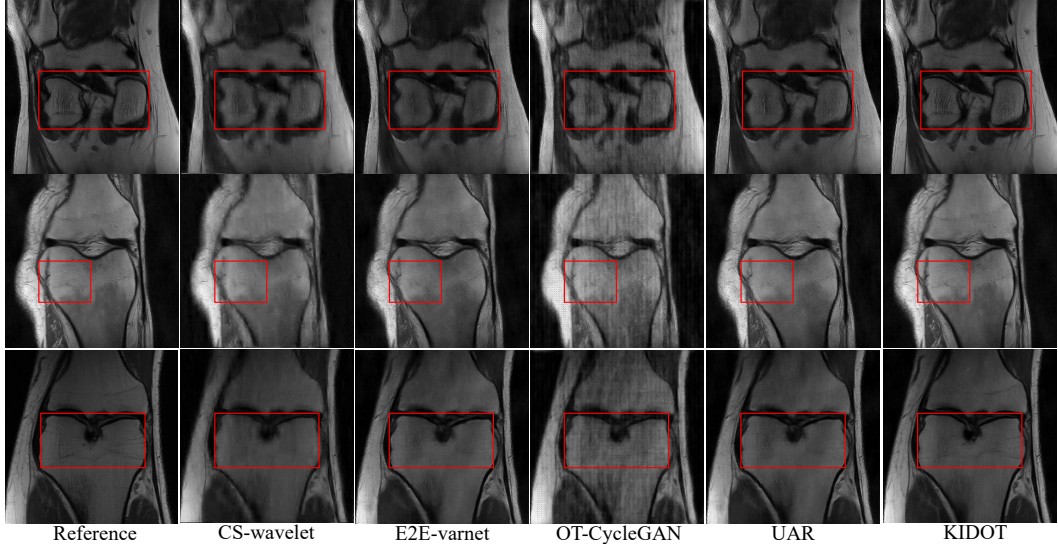

Reference    CS-wavelet    E2E-varnet    OT-CycleGAN    UAR    KIDOT

Figure 8: Visual comparison on Simulated MRI data.

### B.8.2 Additional Visual Comparison on Real Prospective MRI data

Figures 9 and 10 display more visual results of the compared methods.

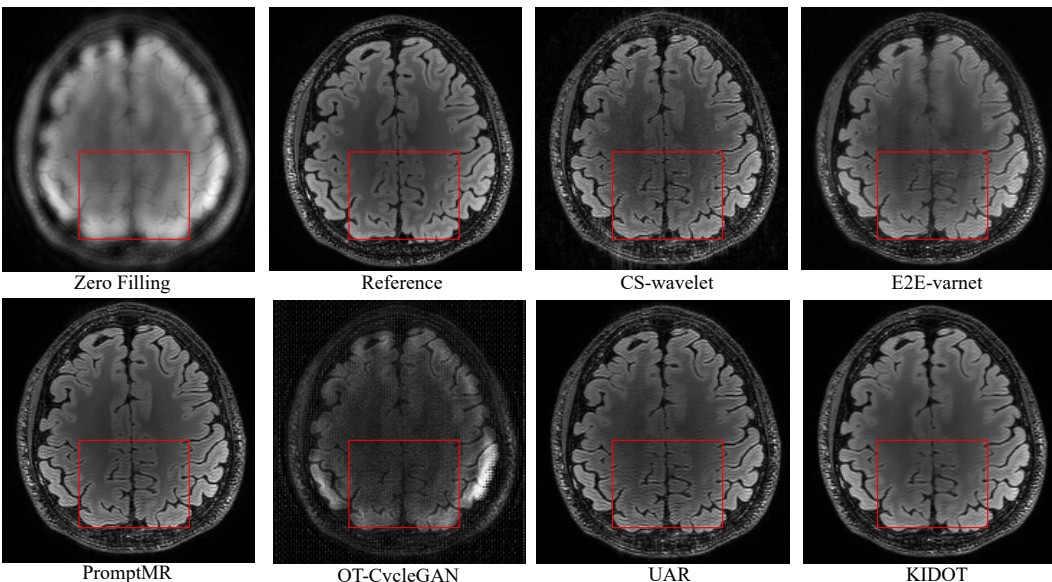

Zero Filling    Reference    CS-wavelet    E2E-varnet

PromptMR    OT-CycleGAN    UAR    KIDOT

Figure 9: Visual comparison on Real prospective MRI data.

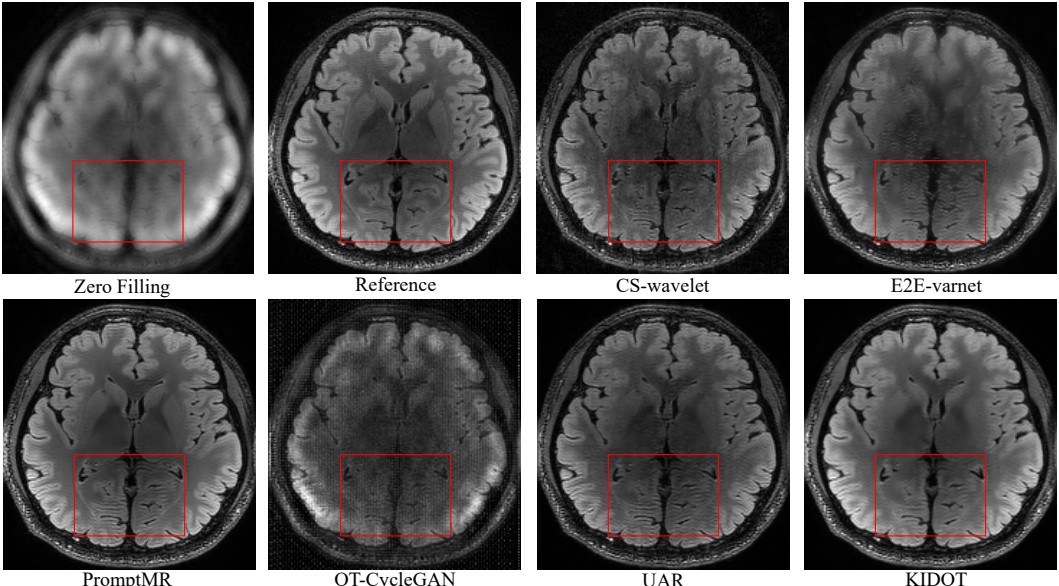

Figure 10: Visual comparison on Real prospective MRI data.

### B.8.3 Additional Visual Comparison on Clinical Prospective CT Data

Figures 11 and 12 displays additional visual results of the compared methods on the clinical prospective CT data. As shown in the regions of interest, our method (KIDOT) achieves a better trade-off between noise suppression and structural detail preservation.

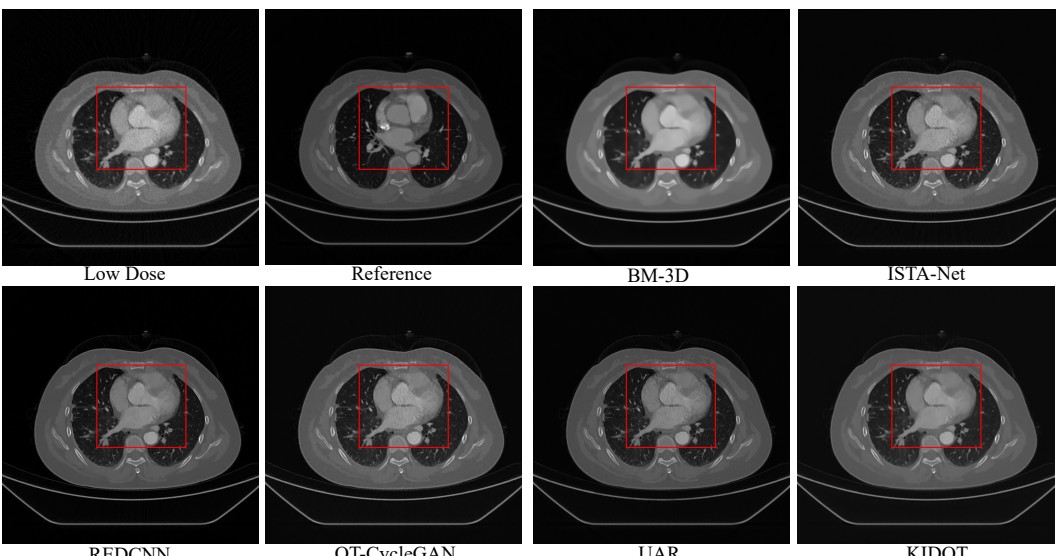

Figure 11: Visual comparison on Clinical prospective CT data.

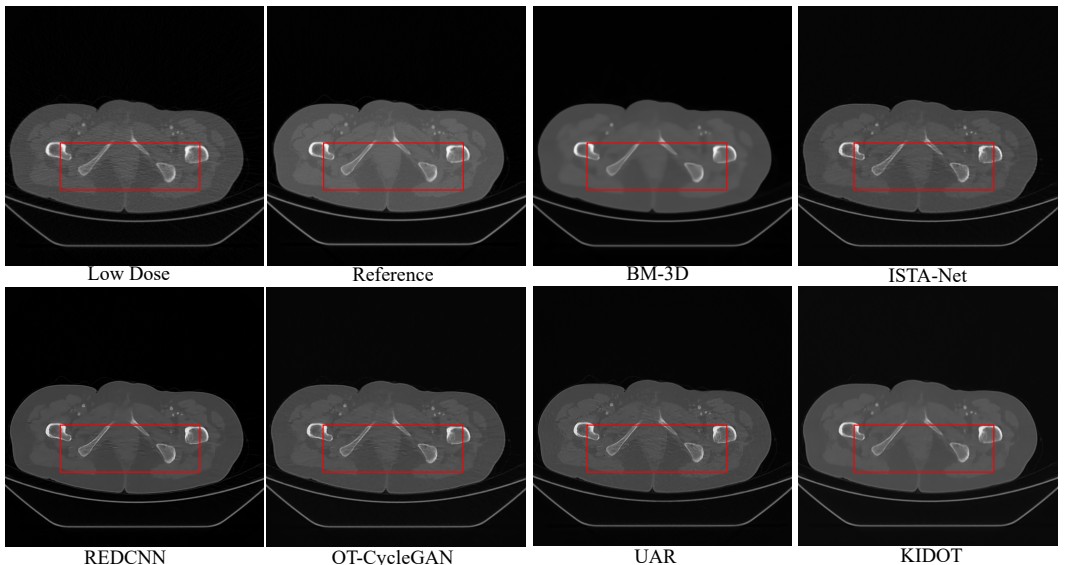

Figure 12: Visual comparison on Clinical prospective CT data.

