# OpenReview forum: "Towards Prospective Medical Image Reconstruction via Knowledge-Informed Dynamic Optimal Transport"
_NeurIPS.cc/2025/Conference — NeurIPS 2025 poster_

### Official Review · Reviewer_JZ2H · 2025-06-08

**Clarity:** 4
**Significance:** 4
**Originality:** 4
**Rating:** 5
**Confidence:** 4

**Summary:**

This paper proposes a new framework for prospective medical image reconstruction. It frames reconstruction as a dynamic transport process, and its learned path is explicitly constrained by imaging physics. Theoretical analysis is performed to validate the existence and optimality of the solution. Experiments on simulated and real MRI/CT datasets demonstrate the effectiveness of the proposed method.

**Questions:**

In Table 4, what is the difference between the last two rows, namely (3) and (4). This is a little confusing. Please make it more clear.

**Ethical Concerns:**

["NO or VERY MINOR ethics concerns only"]

**Final Justification:**

The proposed medical image reconstruction framework is justified by sufficient theoretical analysis and experimental evaluation. The rebuttal also addressed my concerns. Hence I recommend to accept this manuscript.

**Limitations:**

Yes.

**Paper Formatting Concerns:**

N/A.

**Quality:**

4

**Strengths And Weaknesses:**

Strengths:
1. The formulation of prospective medical image reconstruction based on dynamic transport process is reasonable and intriguing.
2. The proposed framework is supported by sufficient theoretical analysis, which builds a solid basis for following development.
3. The proposed method is compared with existing methods on simulated and real MRI/CT datasets, showing the effectiveness of the proposed method.
4. The authors promise the open access to the real prospective datasets, which can benefit the medical imaging community.
5. The paper organization is clear.

Weaknesses:
My concern is mainly about the transport network implementation. The authors use different implementations for the transport network on different tasks, which might be unfair to other comparing methods to some extent. What if using a fixed implementation of the transport network?

---

> ### Author Rebuttal · Authors · 2025-07-31
>
> We thank the reviewer for the insightful comments and suggestions on our paper. We address the concerns and questions in detail below.
>
> ### **W1: Motivation and fairness of transport network implementation**
>
> Thanks for this suggestive question. We clarify that our motivation for network implementation is to use the task-specific and state-of-the-art backbones for each task (e.g., E2E-VarNet for simulated MRI, PromptMR for prospective MRI, ISTA-Net for CT), meanwhile ensuring fairness. We elaborate the motivation as follows.
>
> **Using backbones incorporated with task-specific knowledge:**  In medical image reconstruction, different imaging modalities, such as MRI and CT, possess unique physical properties and data characteristics. For example, MRI heavily relies on multi-coil acquisition, where effectively leveraging the physical information from coil sensitivity maps is often critical for optimal reconstruction. Meanwhile, the imaging mechanism in CT and MRI are different. CT images are produced by Radon Transform and MR images are produced by Fourier Transform. This necessity to incorporate modality-specific knowledge has led the research community to develop specialized, state-of-the-art architectures for each task. To evaluating the performance of our approach for each task, we adopt the state-of-the-art architectures for each task. Note that, for MRI, E2E-VarNet is more light-weight than PromptMR, and we use E2E-VarNet for simulated MRI for the sake of efficient computation.
>
> **Fair comparison in each experiment:** We clarify that our KIDOT and the baselines use the same backbones in each individual experiment. To ensure the comparison is as fair as possible.
>
> **Aiming to demonstrate universal usage of KIDOT on different backbones:** Our another goal is to demonstrate the effectiveness of the proposed KIDOT framework on different network architectures. To isolate the contribution of our framework, we deliberately chose backbones that are identical or highly similar to the state-of-the-art supervised baselines for each respective task. For instance, when comparing with E2E-VarNet on the simulated MRI dataset, our KIDOT model also uses an E2E-VarNet-based architecture as its transport network $T\_\varphi$. This ensures that the performance gain is attributable to our novel knowledge-informed dynamic optimal transport formulation, but the more powerful backbone. The experiments in Tables 1-3 indeed shows the effectiveness of our method on different backbones.
>
> We do believe it is interesting to investigate the universal network architecture for different the medical image reconstruction tasks along with the implementation of KIDOT. Such a universal network architecture cloud be possible if introducing in the attention mechanism, gating mechanism, and mixture of experts model for dealing task-specific and common knowledge, similar to large language models. Due to the limited time of the rebuttal, we left this study as future work.
>
> We will add a detailed explanation in the **"Implementation Details" section (Sec B.1)** of the appendix, explicitly stating why different backbones were chosen and how this contributes to a fair comparison.
>
> ### **Q1: Clarification of Table 4**
>
> We apologize for the confusion and will revise the text to make the distinction unambiguous. We denote $C\_{\rm inter} = \sum\_{i=0}^{N-2} \|y - \mathcal{A}(\mathcal{T}\_\phi^{t_i}(y))\|\_1$, $C\_{\rm final} = \|y - \mathcal{A}(\mathcal{T}\_\phi^{t\_{N-1}}(y))\|\_1$.
>
> Then the key difference between row (3) and row (4) lies in **when and how the imaging-knowledge-informed cost is applied**.
>
> *   **Row (3)  ( $\mathcal{L}\_{\rm SUP}+\varphi\_\theta+ C\_{\rm final}$ ):** This setting represents an ablation where the physics-consistency cost, $\|y - \mathcal{A}(\mathcal{T}\_\phi^{t\_{N-1}}(y))\|\_1$, is applied only to the final output of the N-step transport path. It checks if simply adding a physics-based penalty at the end is beneficial.
> *   **Row (4) (KIDOT= $\mathcal{L}\_{\rm SUP}+\varphi\_\theta+ C\_{\rm inter} + C\_{\rm final}$ ):** This is our full, proposed model. It enforces physics consistency throughout the entire dynamic transport process. As defined in our objective function (Eq. 10), the cost is integrated over all intermediate steps of the path ($\sum\_{i=0}^{N-1} \|y - \mathcal{A}(\mathcal{T}\_\phi^{t_i}(y))\|\_1$).
>
> This comparison is crucial as it demonstrates that enforcing consistency along the entire trajectory (our full KIDOT model) is superior to only constraining the final endpoint, which is a core tenet of our dynamic approach.
>
> We will revise the **"Ablation Study" section (Sec. 4)** and the **method descriptions in Table 4** to explicitly and clearly state this difference.

---

> ### Comment · Reviewer_JZ2H · 2025-08-05
>
> Thank the authors for the rebuttal! It has addressed my previous questions. The authors also promised to provide modifications in the final version. Hence I will keep my positive rating.

---

> > ### Author Response · Authors · 2025-08-05
> > **Thanks to  Reviewer JZ2H**
> >
> > We thank the reviewer again for the valuable review, which enables us to enhance the paper. As suggested, we will carefully revise the paper according to the discussion.

---

### Official Review · Reviewer_kNxi · 2025-07-02

**Clarity:** 3
**Significance:** 3
**Originality:** 4
**Rating:** 5
**Confidence:** 5

**Summary:**

This paper introduces KIDOT, a novel framework for prospective medical image reconstruction. The core idea is to frame reconstruction as a dynamic optimal transport (OT) problem, where the evolution from a degraded measurement to a high-quality image is modeled as a continuous path. Crucially, this transport path is guided by imaging physics, incorporated into both a novel instantaneous cost function and the transport dynamics equation itself. Experiments demonstrate its superiority on simulated and real-world prospective MRI and CT data.

**Questions:**

1. The authors performed experiments on real-world CT datasets and described their data collection process. Will these datasets be made publicly available? If not, are there any existing public datasets that can be used to further evaluate the method to improve reproducibility?

**Ethical Concerns:**

["NO or VERY MINOR ethics concerns only"]

**Final Justification:**

The paper proposes a novel and well-motivated approach, Knowledge-Informed Dynamic Optimal Transport (KIDOT), for prospective medical image reconstruction, which addresses an important practical problem with a principled, physics-grounded method. Extensive experiments, including real clinical data, support the effectiveness of the approach. The authors’ rebuttal satisfactorily addressed my concerns regarding experimental validation and provided further comparative analysis. While a few minor issues remain, they do not affect the overall strength of the contributions. Based on these points, I have increased my score and recommend acceptance.

**Limitations:**

Yes.

**Paper Formatting Concerns:**

No.

**Quality:**

3

**Strengths And Weaknesses:**

Strengths:
1. This paper is well-motivated and tackles a critical and practical problem in prospective medical image reconstruction. The proposed method, Knowledge-Informed Dynamic Optimal Transport (KIDOT), is novel and logically sound. It offers a more principled, physics-grounded approach.
2. This paper clearly introduces and validates the proposed theory, grounding the method in the theory of dynamic optimal transport. The formulation of the imaging-informed cost function and the physics-guided transport equation is elegant.
3. Extensive experiments are conducted, and the proposed method demonstrates state-of-the-art performance. The ablation studies are comprehensive and effectively validate the contribution of their designs.
4. This paper validates the proposed method on both simulated and real, prospectively acquired MRI and CT datasets, which demonstrates the robustness and practical relevance of the method.
5.  This paper is very well-written. The motivation is clearly articulated, and the complex methodology is broken down into understandable components.

Weaknesses:
1. The authors do not explain how the number of discrete time steps is determined for the experiments. And what is the impact on the performance when N increases? Is there a trade-off between the number of steps and reconstruction quality?
2. Although the proposed method achieves the best performance among the compared baselines, most of the compared methods are relatively outdated. It would be beneficial to include comparisons with recent optimal transport-based methods, such as Feedback Schrödinger Bridge Matching.
3. A discussion comparing the strengths and weaknesses of the proposed method relative to recent optimal transport-based methods would further clarify its advantages and limitations, such as the high relevant FSBM.

---

> ### Author Rebuttal · Authors · 2025-07-31
>
> We thank the reviewer for the insightful comments and suggestions on our paper. We address the concerns and questions in detail below.
>
> ### **W1: Determination and impact of the number of discrete time steps ($N$)**
>
> In our work, the number of steps $N$ was treated as a hyperparameter. It was determined through a search on a validation set, with the objective of finding the optimal balance between the fidelity of the ODE approximation and the associated computational cost.
>
> The choice of $N$ represents a critical trade-off. Smaller $N$ can lead to a coarse and inaccurate discretization of the continuous transport path, resulting in suboptimal reconstruction quality. Too large $N$ may increase the computational burden (inference time and memory grow linearly with $N$). Meanwhile, larger $N$ can also potentially lead to instability of training, because the possibility of gradient exploding/vanishing in backpropagation increases for larger $N$. We report the results on the real prospective MRI data (10× acceleration) with varying $N$ as follows:
>
>
> Table r2-1: Ablation study on the number of discretization steps *N*.
>
> | discretization steps *N* | FID ↓ | KID×100 ↓ |
> |:------------------------:|:-----:|:---------:|
> | 6                        | 35.31 | 2.01      |
> | 9                        | 30.06 | 1.62      |
> | **12**                   | **28.26** | **1.12**  |
> | 15                       | 29.23 | 1.31      |
>
> As shown in Table r2-1, the reconstruction quality, measured by both FID and KID, consistently improves as $N$ increases from 6 to 12. Our method achieves the best results when $N=12$. When we further increase the number of steps to 15, the performance begins to degrade, which may be due to the increased possibility of gradient exploding/vanishing in training.
>
>
> We will add this ablation study and the accompanying discussion to the Appendix.
>
>
> ### **W2 & W3: Comparison with recent optimal transport-based methods of FSBM**
>
> We first compare the methodology of FSBM and KIDOT, and then perform experimental comparison of them.
>
> **1. Methodology comparison of FSBM and KIDOT**
>
> **Summary:**
> FSBM tackles the general semi-supervised generative modeling where datasets are partially aligned. It leverages a small subset of pre-aligned "key-points" provide feedback and guide the transport path for unpaired data. FSBM utilizes a distance-preserving objective to build a dynamic transport process in Schrödinger bridge framework.
>
> KIDOT aims at tackling the prospective medical image reconstruction problem. KIDOT models the medical image reconstruction problem as an dynamic transport problem from the low-quality images to the high-quality images. KIDOT explicitly embeds the medical imaging knowledge into the transport cost and equation, forming an new knowledge-informed dynamic optimal transport framework.
>
> **Difference:**
> Though both FSBM and our proposed KIDOT tackles data distribution transform using dynamic optimal transport, they are different in both framework and techniques. The core of FSBM is to guide the translation of unpaired data by the paired key-points using an developed feedback model. While our KIDOT aims to incorporate the imaging knowledge into the transport model by designing the knowledge-informed transport cost and equation, for prospective medical image reconstruction.
>
> **Advantages and limitations of KIDOT:**
> As analyzed above, when the imaging knowledge is clear, our KIDOT has advantage for the task (as verified in experiments), because the imaging knowledge provides inductive bias that can be utilized in KIDOT. However, when the knowledge is not clear or mismatched, the performance of KIDOT could be limited.
>
> **2. Experimental comparison**
>
> To provide a comparison, we implement FSBM and evaluate it across all three of our experimental datasets. Following the setup in the FSBM paper for high-dimensional data, we perform the transport in the latent space of an autoencoder. For MRI experiments, we utilize the publicly available microsoft/mri-autoencoder-v0.1 model on Hugging Face. For CT experiments, we train a new autoencoder from scratch. This model is trained on our CT dataset, ensuring the learned latent representations are optimally tailored to the target domain. To provide FSBM with the necessary 'key-point' pairs for its semi-supervised guidance, we use our simulated paired data as the ground truth. The trajectories for these key-points were generated via linear interpolation between the paired samples, providing the guidance FSBM requires.
>
> Table r2-2: Performance comparison on simulated MRI data
> | Method | PSNR (↑) | SSIM (↑) | FID (↓) | KID*100 (↓) |
> |:--- |:---:|:---:|:---:|:---:|
> | FSBM | 26.52 | 0.7634 | 113.62 | 6.03 |
> | **KIDOT (Ours)** | **33.31** | **0.8518** | **10.68** | **0.83** |
>
> Table r2-3: Performance on real prospective MRI data
> | Method | FID (↓) | KID*100 (↓) |
> |:--- |:---:|:---:|
> | FSBM | 147.62 | 7.13 |
> | **KIDOT (Ours)** | **28.26** | **1.12** |
>
> Table r2-4: Performance on clinical prospective CT data
> | Method | FID (↓) | KID*100 (↓) |
> |:--- |:---:|:---:|
> | FSBM | 68.42 | 3.61 |
> | **KIDOT (Ours)** | **22.25** | **1.32** |
>
> Table r2-2, Table r2-3 and Table r2-4 show the results. The observed difference in performance can be attributed to KIDOT's explicit integration of the physical model. This provides a strong, task-specific inductive bias that directly constrains the reconstruction to be consistent with the measurement $y$. In contrast, FSBM, as a general-purpose generative model, does not have an inherent mechanism to enforce this specific physical constraint, making it less suited for the high-fidelity demands of this inverse problem.
>
> We will incorporate this comprehensive comparison—including the conceptual distinctions and the new experimental results—into the **Related Work** and **Experiments** sections of our paper.
>
> ### **Q1: Data and code availability for reproducibility**
>
> The clinical prospective CT dataset was acquired under a strict Institutional Review Board (IRB) protocol, which places necessary restrictions on the public release of the full patient cohort to protect privacy. However, we recognize the immense value of providing real-world data to the community. Therefore, to the fullest extent permitted by our IRB, **we will release a representative, fully anonymized subset of this prospective CT dataset.** This subset will contain low-dose and standard-dose scans and will enable other researchers to directly evaluate their methods on the same challenging, real-world data distribution we addressed.
>
> We will also release our source code when the paper is public to ensure reproducibility. We will also add a statement to our paper clarifying these points.

---

> > ### Comment · Reviewer_kNxi · 2025-08-06
> >
> > Thank you for thoroughly responding to my comment! Please ensure that these new discussions (experimental results and key discussions) are included in the next version of the paper. I am pleased to raise my score based on these clarifications.

---

> ### Author Response · Authors · 2025-08-06
> **Kindly reminding discussion deadline**
>
> Dear Reviewer kNxi,
>
> We would like to thank you again for the valuable comments and suggestions. We have responded to the comments in detail and hope to address your concerns. As the discussion period is drawing to a close, we kindly remind you that if you have other concerns, please let us know. We will try our best to clarify further.
>
> Best regards,
>
> The Authors

---

> ### Author Response · Authors · 2025-08-06
> **Thanks to Reviewer kNxi**
>
> We thank the reviewer for the valuable feedback. We will carefully revise the paper according to the discussion.

---

### Official Review · Reviewer_YYCZ · 2025-07-02

**Clarity:** 3
**Significance:** 3
**Originality:** 2
**Rating:** 4
**Confidence:** 4

**Summary:**

This paper introduces KIDOT, a dynamic optimal transport framework for medical image reconstruction that integrates imaging physics directly into both the cost function and transport dynamics. The method is designed to work with unpaired data and aims to address the retrospective-to-prospective domain gap, with experiments on MRI and CT showing improved perceptual and fidelity metrics.

**Questions:**

1.	*Regarding weakness 1:* Can the authors provide proof for KIDOT’s superior performance in low-data regimes?
2.	*Regarding weakness 2:* Prospective image for simulated data was generated by flipping 3% of the entries in the training mask, essentially just changing the undersampling pattern slightly. Diffusion-based reconstruction methods, which are typically trained independently of the sampling mask, are known to be inherently robust to such variations. Would KIDOT still provide an advantage over these models under a more meaningful shift, and why are they not included in the comparison?
3.	*Regarding weakness 2:* Would a diffusion model trained on a large existing dataset, such as DDS [1] trained on fastMRI brain data, perform competitively on this prospective dataset at test time? If so, this would suggest that we can still collect prospective data in an unpaired fashion and reconstruct it using strong pre-trained priors, without the need for a method like KIDOT.  Exploring this would help clarify whether KIDOT provides a real advantage in clinical settings.
4.	*Regarding weakness 3:* Unpaired training is often motivated by the ability to use heterogeneous data sources, so how would KIDOT perform when the source and target domains differ more significantly, for example in contrast type, anatomy, or acquisition settings?

[1] Chung et al., “Decomposed Diffusion Sampler for Accelerating Large-Scale Inverse Problems”, ICLR, 2024.

**Ethical Concerns:**

["NO or VERY MINOR ethics concerns only"]

**Final Justification:**

The paper presents strengthened empirical evidence and addresses key concerns, and contingent on the inclusion of the new comparisons and experiments in the final version, I find it suitable for acceptance.

**Limitations:**

Yes, the authors have addressed the limitations in Section 5.

**Paper Formatting Concerns:**

None.

**Quality:**

3

**Strengths And Weaknesses:**

**Strengths:**

1.	The supervision assumption is less restrictive than fully supervised approaches, as it does not require paired measurement-image data. This aligns well with real-world clinical scenarios, where perfectly aligned scans are often unavailable or impractical to obtain.
2.	Theoretical analysis is provided to support the formulation, including proof of existence of solutions under mild assumptions.
3.	The method was tested in various settings, focusing on both simulated and prospective MRI data, and clinical prospective CT data.

**Weaknesses:**

1.	KIDOT is claimed to work without paired data, but it still requires thousands of training slices based on the numbers reported in the paper. There is no evaluation of performance in low-data regimes, which are often the reality in clinical settings. This weakens the motivation for unpaired learning, especially since large paired datasets (such as fastMRI, mridata.org and others) already exist and are widely used.
2.	While KIDOT is proposed as a solution for unpaired and prospective data, the paper does not compare it to diffusion-based reconstruction methods that use strong priors trained on large datasets like fastMRI. Even though these models are trained with paired data, they are often robust to distribution shifts and can perform well on prospective data if the prior matches the anatomy or modality, which is commonly the case in practice.
3.	Although KIDOT is trained on unpaired data, the paper does not explore how well the model generalizes to more aggressive out-of-distribution (OOD) examples, such as different contrasts, anatomies, or acquisition protocols. Evaluating under such conditions would strengthen the paper’s premise, since the core motivation of KIDOT is to address the retrospective-to-prospective gap, which is itself a form of distribution shift.

---

> ### Author Rebuttal · Authors · 2025-07-31
>
> We thank the reviewer for the insightful comments and suggestions on our paper. We address the concerns and questions in detail below.
> ### **W1 & Q1: Clarification on motivation for unpaired learning and experiments in low-data regimes**
>
> We clarify the "paired data" in medical image reconstruction datasets and the motivation of our method, along with the experiments in low-data regimes, respectively.
>
> **1. Clarification of "paired" data in public datasets**
>
> Though there exist large paired public datasets like fastMRI, the "pairing" in these datasets is mainly synthetic. For example, in fastMRI, the low-quality (under-sampled) k-space data is generated by retrospectively simulating a sampling mask on fully-sampled k-space data. Similarly, in low-dose CT datasets like the Mayo Clinic Challenge, the low-dose images are created by simulating noise onto clean, standard-dose scans. These simulation are hard to fully replicate the complex, non-ideal characteristics of real-world prospective data acquisition, such as subtle patient motion between scans, system hardware instabilities, and specific noise distributions. Deep models trained on these "idealized" simulated data often suffer a significant performance drop when deployed in real clinical workflows.
>
> **2. KIDOT's motivation: leveraging abundant real-world data**
>
> We clarify that the motivation for our KIDOT stems from a pragmatic clinical reality. Specifically, in clinical scenario, acquiring **perfectly registered, truly prospective paired data** (e.g., a low-dose scan and a standard-dose scan of the same patient at the exact same moment and position) is extremely difficult, often ethically questionable, and sometimes physically impossible. In contrast, it is **relatively easy and common** to collect large volumes of **unpaired** clinical data: a vast pool of low-quality scans (e.g., under-sampled MRI, low-dose CT) from some patients, and a separate, large pool of high-quality, fully-sampled scans from other patients.
>
> KIDOT is designed to leverage these two abundant but unpaired real-world data streams, bypassing the need for perfect, one-to-one pairing. KIDOT directly tackles the distribution shift problem by learning from the true data distributions encountered in clinical practice.
>
> **3. Performance in low-data regimes**
>
> We first clarify that the paired training data in simulated datasets, such as fastMRI, is generated by retrospectively applying sampling masks to fully-sampled data. This means that, in principle, the supply of such paired data is abundant, limited only by the availability of fully-sampled scans. As suggested, we also evaluate our method when the source of simulated paired data is intentionally restricted. We conduct a new ablation study on the simulated MRI dataset, in which we train methods on reduced data subsets, 20% and 10% of the original training paired data. The results are presented in Table r1-1 and Table r1-2 below.
>
> Table r1-1: Performance comparison in a low-data regime, using 20% of the original training paired data.
> |Method|PSNR (↑)|SSIM(↑)|FID(↓)|KID*100(↓)|
> |:-|:-:|:-:|:-:|:-:|
> |E2E-VarNet|30.97|0.7978|72.43|4.02|
> |UAR|31.30|0.8054|40.32|2.95|
> | **KIDOT (Ours)**|**31.75**|**0.8131**|**31.61**|**2.32**|
>
> Table r1-2: Performance comparison in a low-data regime, using 10% of the original training paired data.
> |Method|PSNR(↑)|SSIM(↑)|FID(↓)|KID*100(↓)|
> |:-|:-:|:-:|:-:|:-:|
> |E2E-VarNet|29.92|0.7681|92.41|6.07|
> |UAR|30.64|0.7857|87.07|5.68|
> |**KIDOT (Ours)**|**30.88**|**0.7952**|**73.14**|**4.11**|
>
> Tables r1-1/2 implies improved performance of KIDOT over baselines. This performance under data scarcity could be directly attributable to the regularization imposed by KIDOT's physics-informed constraints. Unlike purely data-driven methods that are prone to overfitting on small datasets, our framework's adherence to the physical model provides a robust inductive bias, reducing its dependency on the volume of training data.
>
> We will add this new low-data ablation study to the Appendix. We will also include the discussions into the introduction and motivation sections in the paper.
>
> ### **W2 & Q2 & Q3: Comparison with diffusion-based models**
>
> We are sorry for missing the diffusion-based methods. As suggested, we have conducted experiments on both simulated and real datasets to compare with the diffusion method of Denoising Diffusion Sampler (DDS (Chung et al., ICLR 24)).
>
> **Experiment 1: comparison on simulated data**
>
> To compare with diffusion-based reconstruction method, we first performed an experiment on **simulated fastMRI dataset**. The authors of DDS provide a publicly available model pre-trained on fastMRI, enabling a direct comparison between the two methods under the same setting. Table r1-3 shows KIDOT outperforms the pre-trained DDS model.
>
> Table r1-3: Quantitative comparison against the SOTA baseline DDS on the simulated fastMRI dataset.
> | Method | PSNR (↑) | SSIM (↑) | FID (↓) | KID*100 (↓) |
> |:- |:-:|:-:|:-:|:-:|
> | DDS | 31.27 | 0.8136 | 29.46 | 1.65 |
> | **KIDOT (Ours)** | **33.31** | **0.8518** | **10.68** | **0.83** |
>
> **Experiment 2: comparing on real prospective MRI data**
>
> For the real prospective MRI data, a practical challenge arises when applying the DDS model pre-trained on fastMRI, to our clinical dataset due to a mismatch in image dimensions (square size（320x320）for fastMRI v.s. non-squre size (256x176) for our data). We tackle this by padding zeros. Another challenge is the inherent **distribution gap** between the fastMRI data and our real prospective MRI data, which necessitates a more nuanced evaluation. The existence of this gap makes it uncertain whether a pre-trained prior will be effective at all.  Therefore, to ensure a fair and physically valid evaluation, we adopted three practical strategies for comparison:
>
> 1.  **DDS (Pre-trained):** The pre-trained DDS model was used directly. Our `256x176` images were zero-padded to `256x256`, processed, and then cropped back.
> 2.  **DDS (Fine-tuned):** Starting with the pre-trained model used in (1), we then performed fine-tuning on our clinical dataset.
> 3.  **DDS (Trained from scratch):** The DDS architecture was trained from the ground up on our clinical dataset.
>
> Table r1-4: Performance comparison of KIDOT against different application strategies for DDS on the real prospective MRI dataset.
> |Method|FID(↓)|KID*100(↓)|
> |:---|:---:|:---:|
> |DDS(Pre-trained)|62.94|2.71|
> |DDS(Fine-tuned)|39.78|1.84|
> |DDS(Trained from scratch)|58.32|2.53|
> |**KIDOT (Ours)**|**28.26**|**1.12**|
>
> Table r1-4 shows the comparison of KIDOT against different application strategies for DDS on the real prospective MRI dataset. The performance of the DDS (Pre-trained) confirms the existence of the gap between the fastMRI data and our real prospective MRI data. DDS (Fine-tuned) improves performance, but it still lags behind KIDOT. Interestingly, DDS (Trained from scratch) on this dataset performs even worse, suggesting that our authentic clinical dataset may be insufficient for a data-hungry model like DDS to learn a robust prior from scratch. KIDOT achieves superior performance. Its physics-guided dynamics provide a powerful inductive bias that may be both more data-efficient and better aligned with the reconstruction task.
>
> We will add these experiments into Tables 1-2 and the detailed analysis into the Experiments section of the paper.
>
> ### **W3 & Q4: Generalization to more aggressive OOD or  heterogeneous data**
>
> As suggested, we investigate the performance on cross-anatomy and cross-contrast experiments on the fastMRI dataset.
>
> *   **Cross-Anatomy (Knee → Brain):** In this setup, models are trained on the knee anatomy domain. Paired baselines utilize paired knee scans for training. Our unpaired method is trained using unpaired high-quality **knee scans** and low-quality **brain scans**. The evaluation is then performed on the task of reconstructing **brain scans**.
> *   **Cross-Contrast (T1 → T2):** Similarly, for this task, models are trained on a dataset of unpaired high-quality **T1W brain scans** and low-quality **T2W brain scans**. We then evaluate their generalization performance on **T2W brain scans**.
>
> The results from these experiments, presented in Table r1-5 and Table r1-6, provide quantitative evidence for KIDOT's physics-guided inductive bias. This bias encourages the model to learn a generalizable reconstruction process grounded in the principles of MR imaging, rather than simply memorizing the anatomical features of the training domain. We anticipated that a standard supervised baseline, while performant on in-domain data, would struggle when faced with a shift in anatomy, as its learned prior is purely data-driven.
>
> Table r1-5: Generalization performance on the Cross-Anatomy (Knee → Brain) task.
> | Method | PSNR (↑) | SSIM (↑) | FID (↓) | KID*100 (↓) |
> |:- |:-:|:-:|:-:|:-:|
> | E2E-VarNet | 33.82 | 0.7880 | 80.21 | 4.24 |
> | **KIDOT (Ours)** | **34.62** | **0.8145** | **26.87** | **1.53** |
>
> Table r1-6: Generalization performance on the Cross-Contrast (T1 → T2) task.
> | Method | PSNR (↑) | SSIM (↑) | FID (↓) | KID*100 (↓) |
> |:- |:-:|:-:|:-:|:-:|
> | E2E-VarNet | 32.06 | 0.8229 | 31.29 | 1.42 |
> | **KIDOT (Ours)** | **32.48** | **0.8348** | **29.74** | **1.33** |
>
> Due to space limit, we will include this experiment to Appendix.

---

> > ### Comment · Reviewer_YYCZ · 2025-08-05
> > **Official Comment by Reviewer YYCZ**
> >
> > I would like to thank the authors for their clear, detailed, and well-structured rebuttal, which effectively addressed my concerns. I appreciate the clarifications, and I believe the added comparisons to diffusion-based baselines, as well as the new experiments on cross-contrast and low-data regimes, significantly strengthen the paper. I look forward to seeing these additions reflected in the final version. Accordingly, I have raised my score, contingent on their incorporation, as I believe the paper would benefit the community and merits acceptance.

---

> > > ### Author Response · Authors · 2025-08-05
> > > **Thanks to Reviewer YYCZ**
> > >
> > > We thank the reviewer for the valuable feedback, and the insightful comments/suggestions. We will revise the paper accordingly, as discussed in the response.

---

### Decision · Program_Chairs · 2025-09-17

**Decision:**

Accept (poster)

**Comment:**

The paper introduces a novel framework for prospective medical image reconstruction that leverages optimal transport and imagining physics. The method is validated on simulated and real-world prospective MRI and CT data.

Initially, the reviewers agreed on the importance of the tackled problem, the novelty of the approach and praised the good writing of the paper. Most of the initial reviewer’s questions revolved around experimental setups and additional comparisons. In the rebuttal, the authors clarified the experimental setup and provided additional evaluations. In the discussion period all reviewers were satisfied with the authors’ responses and recommended the acceptance of the paper. AC agrees with the reviewers and recommends to accept.